

# Accounting for Carbon and Nitrogen interactions in the Global Terrestrial Ecosystem Model ORCHIDEE (trunk version, rev 4999): multi-scale evaluation of gross primary production

Nicolas Vuichard[1], Palmira Messina[1,+], Sebastiaan Luyssaert[2], Bertrand Guenet[1], Sönke Zaehle[3], Josefine Ghattas[4], Vladislav Bastrikov[1], Philippe Peylin[1]

[1]Laboratoire des Sciences du Climat et de l'Environnement, LSCE/IPSL, CEA-CNRS-UVSQ, Université Paris-Saclay, 91191 Gif-sur-Yvette, France
[2]Vrije Universiteit Amsterdam, Faculty of Science, 1081 HV, the Netherlands
[3]Max Planck Institute for Biogeochemistry, Hans-Knöll-Str. 10, 07745 Jena, Germany
[4]Institut Pierre-Simon Laplace, IPSL, UPMC 4 Place Jussieu, 75252 Paris, France
[+]Now at Centre d'Enseignement et de Recherche en Environnement Atmosphérique, CEREA/IPSL, 77455 Marne la Vallée, France

*Correspondence to*: Nicolas Vuichard (vuichard@lsce.ipsl.fr)

**Abstract.** Nitrogen is an essential element controlling ecosystem carbon (C) productivity and its response to climate change and atmospheric $[CO_2]$ increase. This study presents the evaluation – focussing on gross primary production (GPP) – of a new version of the ORCHIDEE model that gathers the representation of the nitrogen cycle and of its interactions with the carbon cycle from the OCN model and the most recent developments from the ORCHIDEE trunk version.

We quantify the model skills at 78 Fluxnet sites by simulating the observed mean seasonal cycle, daily mean flux variations, and annual mean average GPP flux for grasslands and forests. Accounting for carbon-nitrogen interactions does not substantially change the main skills of ORCHIDEE, except for the site-to-site annual mean GPP variations, for which the version with carbon-nitrogen interactions is in better agreement to observations. However, the simulated GPP response to idealized $[CO_2]$ enrichment simulations is highly sensitive to whether or not carbon-nitrogen interactions are accounted for. Doubling of the atmospheric $[CO_2]$ induces an increase of the GPP, but the site-averaged GPP response to $CO_2$ increase projected by the model version with carbon-nitrogen interactions is half of the increase projected by the version without carbon-nitrogen interactions. This model's differentiated response has important consequences for the transpiration rate, which is on average 50 mm $yr^{-1}$ lower with the version with carbon-nitrogen interactions.

Simulated annual GPP for northern, tropical and southern latitudes shows good agreement with the observation-based MTE-GPP product for present-day conditions. An attribution experiment making use of this new version of ORCHIDEE for the time period 1860-2016 suggests that global GPP has increased by 50%, the main driver being the enrichment of land in reactive nitrogen (through deposition and fertilization), followed by the $[CO_2]$ increase.

Based on our factorial experiment and sensitivity analysis, we conclude that if carbon-nitrogen interactions are accounted for, the functional responses of ORCHIDEE r4999 better agrees with current understanding of photosynthesis than when the



carbon-nitrogen interactions are not accounted for, and that carbon-nitrogen interactions are essential in understanding global terrestrial ecosystem productivity.

## 1 Introduction

Global Terrestrial Ecosystem Models (GTEMs) are mathematical models which are dedicated to provide a better
understanding of terrestrial ecosystem functioning and its interplay with environmental drivers such as temperature or precipitation. GTEMs aim at simulating the spatial patterns of the fluxes of carbon, water and energy between the land surface and the atmosphere, and their time evolution, in particular in a context of climate change. For over a decade, GTEMs account for climate forcing as well as the effect of atmospheric $CO_2$ concentration (atmospheric $[CO_2]$) on ecosystem productivity (Melillo et al., 1995). Atmospheric $[CO_2]$ is a key driver of the assimilation of carbon by photosynthesis. While
the current atmospheric $[CO_2]$ value is nearby the optimal value for carbon assimilated by plants with a C4 photosynthetic pathway, it is still suboptimal for plants with a C3 pathway (Pearcy and Ehleringer, 1984). In the context of global change where atmospheric $[CO_2]$ is increasing, quantifying the so-called $[CO_2]$ fertilization effect, i.e., the increase in ecosystem productivity associated to increasing atmospheric $[CO_2]$ has been on the forefront (Lobell and Field, 2008; Wullschleger, S. D., Post, W. M., & King, 1995).

Most of the GTEMs estimate a large global land carbon sink over the 21[st] century when used in Earth System Models for climate predictions (of the order of 120-270 GtC over the period 2010-2100, depending on the Representative Concentration Pathways), mainly driven by atmospheric $[CO_2]$ increase (Ciais et al., 2013). Even if atmospheric $[CO_2]$ will be plentiful, it remains questionable whether sufficient nutrients, in particular nitrogen, will be available to fully sustain the associated
increase of primary production. A recent study (Zaehle et al., 2015) estimated that 40 to 80% of the carbon sequestration on land projected by simulations without nutrient limitations for the period 1851-2100, will not occur if nitrogen limitation and carbon-nitrogen interactions were accounted for in GTEMs embedded into Earth system models. The 40% variation in the projected N-limitation of the land carbon sink was reported to depend on the evolution of the anthropogenic production of reactive nitrogen and associated atmospheric nitrogen deposition, which differ for each Representative Concentration
Pathway (Ciais et al., 2013).

Only two GTEMs involved in the last exercise of the Climate Modelling Intercomparison Project (CMIP5) accounted for carbon-nitrogen interactions (CESM1-BGC and NorESM1-M). Since then many GTEMs have been further developed to account for the impact of the nitrogen cycle (Churkina et al., 2009; Esser et al., 2011; Fisher et al., 2010; Goll et al., 2017;
Jain et al., 2009; Smith et al., 2014; Sokolov et al., 2008; Thornton et al., 2007; Wang et al., 2007, 2010; Xu-Ri and Prentice, 2008; Zaehle and Friend, 2010), some of which being included in an Earth system model. Among the GTEMs that developed carbon-nitrogen interactions was an ORCHIDEE-derived model named OCN (Zaehle and Friend, 2010), that has been used



in several studies (Zaehle et al., 2010b, 2011, 2014). However, this pionneering development (2007-2010) has not been embedded in subsequent versions of the ORCHIDEE model, especially with respect to the coupling to the French IPSL (Institut Pierre Simon Laplace) Earth system model. The present paper presents and evaluates a recent modelling effort consisting of the merge of the ORCHIDEE trunk version (r3977) with the carbon-nitrogen interactions based on Zaehle and

Friend (2010). It describes all changes made in the original nitrogen cycle and carbon-nitrogen interactions, linked to major updates in the water, carbon and energy budgets in ORCHIDEE since the first development of OCN, together with a thourough evaluation of simulated gross carbon uptake and transpiration by plants.

## 2 Methods

Following a description of the model developments, the evaluation is focused on the carbon cycle and on the added value of

including the nitrogen cycle for the purpose of simulating gross carbon uptake fluxes, including also the impact on the related plant transpiration. The evaluation consists of: (1) site-level simulations in order to assess the overall performance of the ORCHIDEE model at simulating GPP flux at Fluxnet stations (annual mean value, seasonal variations, site-to-site differences); (2) sensitivity tests to quantify the contributions of accounting for seasonal and site-to-site variations of the leaf C/N ratio to the simulated seasonal variations and mean annual GPP; (3) idealized simulations to quantify the impact of N-

limitation on GPP under [$CO_2$] enrichment scenarios; and (4) global simulations in order to evaluate and analyse seasonal, long-term variations and global distribution of the simulated GPP.

### 2.1 Model description

This section focuses on the major modifications that were included in the ORCHIDEE model since the first implementation of a nitrogen cycle in the OCN model (Zaehle and Friend, 2010). The ORCHIDEE model calculates the exchange of energy,

water, carbon and nitrogen at the atmosphere-surface interface and within the soil-plant continuum. The main modelling structure originates from Ducoudré et al., (1993) and Krinner et al., (2005) for water- and energy-related and carbon-related processes, respectively. The spatial discretization depends on the modelled process. The energy budget is computed at the grid cell level, without accounting for differences between tiles within the grid cell. Water budgets are now calculated for three tiles per grid cell, one for the bare soil, one for tree cover and one for herbaceous cover. The carbon budget and related

fluxes are computed for each vegetated tile within a grid cell. In ORCHIDEE, the carbon in the vegetation and soil is split in thirteen classes, based on the concept of Plant Functional Type (PFT). Different species, which share similar characteristics regarding their architecture, phenology or location, are gathered in a single PFT. The thirteen PFTs used in ORCHIDEE are listed in Table 1. Compared to the model version that was used as the basis for OCN, a large portion of the code has been revised and new developments added (Peylin et al., in prep.). The main changes consist of :

i.    a multi-layer soil hydrological scheme which accounts for water diffusion and deep drainage, based on the initial work of de Rosnay (2002) that was shown to better simulate soil water dynamics compared to the initial double





buckets scheme as described in Ducoudré et al. (1993). This feature within the current version of ORCHIDEE has been recently evaluated over Amazonia (Guimberteau et al., 2014) and Africa (Traore et al., 2014);

ii. a revised thermodynamic scheme which accounts for the heat transported by liquid water into the soil, in addition to improvements in the representation of heat conduction process (Wang et al., 2016) which resolves former

inconsistencies between the soil water and energy dynamics;

iii. an analytical solution to the set of equations governing the soil organic matter (SOM) pools and their time evolution driven by the litter input and the climate conditions in terms of soil temperature and humidity (Lardy et al., 2011). The latter is now used to determine SOM pools associated to initial conditions and guarantees steady state conditions better than iterative simplified schemes (Krinner et al., 2005).

iv. A revised parameterisation of the vegetation and snow albedo (i.e. optimized parameters using remote sensing albedo data from MODIS sensor);

The developments in ORCHIDEE r4999 that relate to the nitrogen dynamics within the soil-plant-atmosphere continuum and the dependency between carbon and nitrogen cycles, as well as the allocation scheme and the short- and long-term storage

pools dynamic mostly follow the work of Zaehle et al. (2010b) and Zaehle and Friend (2010). The nitrogen cycle is added at the PFT level as for the carbon cycle and for each carbon pool there is a corresponding nitrogen pool, with C/N ratios evolving through time. Leaf C/N ratio is dynamic and varies as a result of the nitrogen supply by roots and demand for biomass allocation (see Sect. 2.1.3 for details). C/N stoichiometry of the other living biomass pools (i.e., below and above-ground sapwood, below and above-ground heartwood, fruit and fine roots) is driven by the C/N ratio of the leaves, but

multiplied by a pool-dependent factor $f_{cn}$ ($f_{cn}$ equals 1.2 for fine roots and fruit, and 11.5 for the other pools). SOM decomposition follows the scheme of Parton et al. (1993) in which C/N ratios of SOM pools are expressed as a function of soil mineral nitrogen content (ammonium and nitrate). This scheme is rather simple and does not account for known processes such as the priming effect (Fontaine et al., 2007). As a consequence, carbon decomposition rates are independent of the C/N ratio of the SOM pools, which facilitate the use of an analytical solution for quantifying the carbon content of

SOM pools at equilibrium.

In ORCHIDEE r4999, the fate of mineral nitrogen in the soil follows the formalism of the OCN model (Zaehle and Friend, 2010) mainly based on the DNDC model (Li et al., 1992, 2000; Zhang et al., 2002). The formalism accounts for ammonium ($NH_3/NH_4^+$), nitrate ($NO_3^-$), nitrogen oxides ($NO_x$) and nitrous oxide ($N_2O$) soil pools and associated emissions due to

nitrification (the oxidation of $NH_3/NH_4^+$ in $NO_3^-$) and denitrification (the reduction of $NO_3^-$ up to the production of $N_2$) processes. $NO_3^-$ (resp. $NH_4^+$) uptake by roots is modelled as a function of the $NO_3^-$ (resp. $NH_4^+$) available in the soil (Kronzucker et al., 1995, 1996) and of the root biomass. The more root biomass or the higher the soil $NO_3^-$ (resp. $NH_4^+$) pool, the higher the $NO_3^-$ (resp. $NH_4^+$) uptake is. Nitrogen inputs in the soil/plant system are related to (i) nitrogen deposition under the form of $NH_x$ and $NO_y$ components; (ii) biological nitrogen fixation and nitrogen fertilisation over managed





grasslands and croplands. The nitrogen output fluxes are associated to runoff and leaching, and emissions of $NH_3$, $NO_x$, $N_2O$ and $N_2$.

The main modifications compared to the work of Zaehle and Friend (2010) are related to the carbon assimilation or photosynthesis scheme (Yin and Struik, 2009) and refinements of the N-dependency of the photosynthetic activity (Kattge et al., 2009). These developments are described in the following sections 2.1.1 and 2.1.2. Furthermore, the present study considers biological nitrogen fixation rates invariant in time and computed them as a function of evapotranspiration based on the work of Cleveland et al. (1999) as is the case for the OCN model. The forcings used for the other nitrogen input fields are detailed in Sect. 2.3.4 and 2.3.5.

**2.1.1 Carbon assimilation scheme**

The updated carbon assimilation scheme used in ORCHIDEE r4999 has been proposed by Yin and Struik (2009). This scheme is based on the model developed by Farquhar, von Caemmerer and Berry in the FvCB model, (Farquhar et al., 1980) which predicts carbon assimilation for C3 plants as the minimum of the Rubisco-limited rate of $CO_2$ assimilation (Ac) and the electron transport-limited rate of $CO_2$ assimilation (Aj). Yin and Struik (Yin and Struik, 2009) propose a C4-equivalent

version of the FvCB model and analytical solutions to the set of equations which link the net assimilation rate, the stomatal conductance, and the intercellular $CO_2$ partial pressure.

ORCHIDEE r4999 retained most of the parameter values of the FvCB model as proposed by Yin and Struik (2009), except the parameters which define the maximum rate of Rubisco activity-limited carboxylation ($Vc_{max}$, μmol $CO_2$ m$^{-2}$[leaf] s$^{-1}$) and

the maximum rate of electron transport (e$^-$) transport under saturated light ($J_{max}$, μmol e- m$^{-2}$[leaf] s$^{-1}$) for C3 plants, which were replaced by the formulation and parameterization proposed by Kattge and Knorr (2007). In ORCHIDEE (r4999), $Vc_{max}$ and $J_{max}$ at temperature $T$, are defined as:

$$k_T = k_{ref} exp\left[H_{a,k}(T_l - T_{ref})/(T_{ref}RT_l)\right] \frac{1+exp\left(\frac{T_{ref}\Delta S_k - H_{d,k}}{T_{ref}R}\right)}{1+exp\left(\frac{T_l\Delta S_k - H_{d,k}}{T_lR}\right)}, \qquad (1)$$

where $k$ is either $Vc_{max}$ or $J_{max}$, $k_{ref}$ is the parameter value at the reference temperature ($T_{ref}$ is set to 25°C, expressed in Kelvin

in Eq. (1)), $T_l$ is the leaf temperature (°K), $R$ is the universal gas constant (8.314 J K$^{-1}$ mol$^{-1}$), $\Delta S_k$ an entropy factor (J K$^{-1}$ mol$^{-1}$), and $H_{a,k}$ and $H_{d,k}$, respectively an energy of activation and deactivation (J mol$^{-1}$). Based on the reanalysis of the temperature dependency of $Vc_{max}$ and $J_{max}$ performed by Kattge and Knorr (2007), $\Delta Sk$ with $k$ being either $Vc_{max}$ or $J_{max}$ acclimate to temperature and consequently are expressed as linear functions of the monthly mean leaf temperature ($t_{growth}$, °C). The ratio of $Vc_{max,ref}$ to $J_{max,ref}$ also acclimates to temperature, and Kattge and Knorr (2007) proposed to define it as a

linear function of $t_{growth}$. Consequently, $J_{max,ref}$ can be expressed as :

$$J_{max,ref} = (a_{rJ,V} + b_{rJ,V} \; t_{growth})Vc_{max,ref} \qquad (2)$$

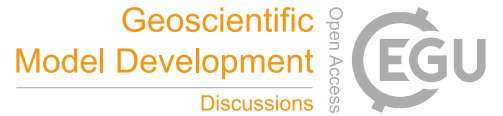



where $a_{rJ,V}$ and $b_{rJ,V}$ are fitted parameters.

### 2.1.2 Nitrogen dependency of photosynthesis activity

ORCHIDEE r4999 accounts for the nitrogen limitation on photosynthetic activity in a different manner than in OCN (Friend and Kiang, 2005; Kull and Kruijt, 1998; Zaehle and Friend, 2010), by making $Vc_{max,ref}$ a function of the leaf nitrogen content

($N_l$, $g_{[N]}$ m$^{-2}$$_{[leaf]}$) as proposed and parameterized by Kattge et al. (2009):

$$Vc_{max,ref} = NUE_{ref}\, N_l \,, \tag{3}$$

where $NUE_{ref}$ is the nitrogen use efficiency (µmol $CO_2$ $g^{-1}_{[N]}$ $s^{-1}$). $NUE_{ref}$ has been widely measured (Ellsworth et al., 2004; Medlyn and Jarvis, 1999; Woodward et al., 1995) and was reported to be PFT-dependent (Kattge et al., 2009) (see Table 1).

Following observations of vertical N-profiles in tree canopies (Thornton and Zimmermann, 2007), $N_l$ is exponentially decreasing from the top to bottom of canopy and its value at a cumulative leaf area index ($L$; m$^2$$_{[leaf]}$ m$^{-2}$$_{[ground]}$, starting from the top of the canopy) is defined following Dewar et al. (2012), with a specific extinction coefficient ($k_N$, value of 0.15) that differs from the one used to calculate the light profile within the canopy (value of 0.5):

$$N_l(L) = N_l(0)\exp(-k_N L) \,, \tag{4}$$

where $N_l(0)$ the value of $N_l$ at top of canopy is expressed as a function of $N_{tot}$ ($g_{[N]}$ m$^{-2}$$_{[ground]}$), the total canopy nitrogen content, and $L_{tot}$ the LAI of the total canopy :

$$N_l(0) = \frac{k_N N_{tot}}{1 - \exp(-k_N L_{tot})} \,, \tag{5}$$

As in Dewar et al. (2012), it is assumed that the variation of the leaf nitrogen content ($N_l$) through the canopy is due to variation of the specific leaf area (SLA, defined as the leaf area divided by the leaf mass , m$^2$$_{[leaf]}$ $g^{-1}_{[C]}$), the leaf nitrogen concentration ($N_{conc}$; $g_{[N]}$ $g^{-1}_{[C]}$) being constant through the canopy. It is also assumed that the SLA at the bottom of the canopy ($sla_{fix}$) is fixed. This implies that the mean SLA of the canopy ($SLA_{mean}$) is no longer a fixed value as was formerly the case in ORCHIDEE, but varies with the total leaf mass ($m_{leaf}$, $g_{[C]}$ m$^{-2}$$_{[ground]}$). $SLA_{mean}$ can be written as:

$$SLA_{mean} = \frac{ln(1 + k_N m_{leaf} sla_{fix})}{k_N m_{leaf}} \,, \tag{6}$$

and $L_{tot}$ as:

$$L_{tot} = SLA_{mean} m_{leaf} \,, \tag{7}$$

### 2.1.3 Nitrogen-related model configurations

Two model configurations were developed to allow a straightforward analysis of the effect of the nitrogen cycle on plant

productivity: one with prescribed leaf nitrogen concentrations and the other with leaf nitrogen concentrations varying



according to nitrogen availability. In the first configuration named "CNfix", the leaf C/N ratio is fixed to a value within a prescribed range ([$CN_{leaf,min}$; $CN_{leaf,max}$], see Table 1). In this configuration the limitation of ecosystem productivity by nitrogen availability is reflected by the imposed leaf C/N ratio, which is fixed for the entire simulation (see Sect. 2.4.3 for the different simulations performed with the "CNfix" configuration). In the "CNfix" configuration, the mass balance of the N

cycle within the soil-plant continuum is closed by taking the nitrogen that is needed for maintaining the imposed leaf C/N ratio from the atmosphere. Rather than implying the absence of nitrogen limitation, the "CNfix" configuration implies a fixed nitrogen limitation, which will not change over time depending on the nitrogen availability. In the other configuration (labelled "CNdyn"), the leaf C/N ratio is not fixed but dynamic. The variation of the leaf C/N ratio ($CN_{leaf}$, $g_{[C]}$ $g_{[N]}^{-1}$) is the outcome of the N-supply from the roots vs. the N-demand to convert the assimilated carbon into leaf, wood, root and fruit

tissue each with its own C/N ratio.

Irrespective of the configuration the model first calculates the nitrogen required ($G_{Ninit}$, $g_{[N]}$ $m^{-2}_{[ground]}$ $d^{-1}$) for satisfying the new carbon allocated to the different reservoirs, $G_C$ ($g_{[C]}$ $m^{-2}_{[ground]}$ $d^{-1}$) under the assumption that $CN_{leaf}$ does not vary (Zaehle and Friend, 2010).

$$G_{Ninit} = \left(f_l/CN_{leaf} + f_r/CN_{root} + f_f/CN_{fruit} + f_s/CN_{sap}\right)G_C \; ,  \qquad (8)$$

where $f_i$ are the fractions (unitless) of carbon allocated to leaf (*l*), roots (*r*), fruit (*f*) and sapwood or stalks (*s*) and $CN_i$ are the C/N ratios (unitless) for the different biomass pools at the previous time step. Assuming that the differences in C/N ratio among the different pools are fixed, they can all be expressed as functions of the C/N ratio of the leaves and the nitrogen

required can be further expressed as:

$$G_{Ninit} = (f_l + \lambda_r f_r + \lambda_f f_f + \lambda_s f_s)/CN_{leaf} \times G_C \; ,  \qquad (9)$$

where $\lambda_i$ are unitless coefficients which reflect the differences in C/N ratio of roots, fruit, and sapwood or stalks, respectively to $CN_{leaf}$. Given that all the available nitrogen is stored in the labile pool ($N_{lab}$, $g_{[N]}$ $m^{-2}$), the model will then check whether

there is sufficient nitrogen in the labile pool to satisfy the demand (if $G_{Ninit} < N_{lab}$) resulting in a decrease of the leaf C/N ratio of the newly allocated biomass ($CN_{leaf,alloc}$) or contrary, that the labile pool does not contain sufficient nitrogen to satisfy the demand  (if $G_{Ninit} > N_{lab}$) resulting in an increase of $CN_{leaf,alloc}$. For this purpose, a dynamic elasticity variable ($D_{leaf}$) is used to dampen the leaf C/N variations and to ensure that $CN_{leaf,alloc}$ remains within the prescribed range of variation, [$CN_{leaf,min}$; $CN_{leaf,max}$]. At each time step, $CN_{leaf,alloc}$ is defined from the value of $CN_{leaf}$, as:

$$CN_{leaf,alloc} = \frac{CN_{leaf}}{D_{leaf}} \; ,  \qquad (10)$$

where




$$D_{leaf} = \begin{cases} max\left(\frac{N_{lab}}{G_{Ninit}}, 1.-0.25 \times D_{max}\right), \; N_{lab} < G_{Ninit} \\ min\left(\frac{N_{lab}}{G_{Ninit}}, 1.25 - 0.25 \times D_{max}\right), \; N_{lab} \geq G_{Ninit} \end{cases}, \tag{11}$$

The elasticity variable $D_{max}$, which avoids rapid changes in $CN_{leaf,alloc}$, was slightly modified compared to its initial implementation in OCN (Zaehle and Friend, 2010) in order to have a value of 1 for $D_{max}$ when $CN_{leaf}$ equals $CN_{leaf,min}$. $D_{max}$

is now calculated as:

$$D_{max} = \exp\left(-\left(a_{D_{max}} \frac{(NC_{leaf,max} - NC_{leaf})}{(NC_{leaf,max} - NC_{leaf,min})}\right)^{b_{D_{max}}}\right), \tag{12}$$

where $NC_{leaf}$, $NC_{leaf,max}$ and $NC_{leaf,min}$ correspond to $1/CN_{leaf}$, $1/CN_{leaf,min}$ and $1/CN_{leaf,max}$, respectively and $a_{D_{max}}$ and $b_{D_{max}}$ are two empirical parameters equal to 1.6 and 4.1, respectively. In the extreme case where $G_{Ninit}$ is greater than $N_{lab}$ while $CN_{leaf}$

equals $CN_{leaf,max}$, the new carbon allocated $G_C$ is lowered in order to maintain $CN_{leaf}$ at the level of $CN_{leaf,max}$.

When $N_{lab}$ is lower than $G_{Ninit}$, the closer $CN_{leaf}$ is to $CN_{leaf,min}$, the higher the nitrogen content reduction of the newly allocated biomass. On the opposite, when $N_{lab}$ is higher than $GN_{init}$, the closer $CN_{leaf}$ is to $CN_{leaf,max}$, the higher the nitrogen content enrichment of the newly allocated biomass.

## 2.2 Evaluation data

### 2.2.1 Fluxnet GPP product

The free fair-use Fluxnet LaThuile collection (http://fluxnet.fluxdata.org/data/la-thuile-dataset/) was used to evaluate the model performance at individual sites. We selected the observation-derived GPP flux based on the NEE partitioning method of Reichstein et al., (2005). From the 153 sites contained in the collection, we selected sites with vegetation belonging to a single PFT, and thus excluded sites covered by a mixture of vegetation (such as savannahs or opened forests). This is to

avoid having to set fractions of PFT, that are usually uncertain, which would introduce substantial uncertainty in the evaluation process. Furthermore, sites were excluded if their vegetation cover was not explicitly represented in ORCHIDEE, such as shrublands or wetlands. In addition, because the nitrogen fertiliser inputs are not reported in the Fluxnet database for managed agro-ecosystems such as grasslands and croplands, which are fertilized with manure or synthetic forms, they were also excluded from the analyses. These two filtering criteria reduced the validation set to 86 sites. Last, we removed eight

sites for which the annual mean precipitation derived from in-situ measurements was highly different of the climatological mean provided by the site principal investigator and of the value derived from the ERA-interim reanalysis (see Vuichard and Papale, 2015). The selected 78 sites (Table A1) were distributed across the following vegetation classes: two Tropical Evergreen Broadleaved Forests sites (TrEBF), 29 Temperate Evergreen Needleleaved Forest (TeENF), seven Temperate Evergreen Broadleaved Forest (TeEBF), 21 Temperate Deciduous Broadleaved Forest (TeDBF), eight Boreal Evergreen

Needleleaved Forest (BoENF) and 11 C3 natural grasslands (GRAc3).



### 2.2.2 Global MTE-GPP product

The observation-based MTE-GPP product (Jung et al., 2011) was used to evaluate the global scale simulations of GPP. MTE-GPP scales up observed half-hourly GPP at Fluxnet stations to global monthly maps at 0.5 x 0.5 degree resolution for the period from 1982 to 2008 based on independent predictors, combined through a machine learning technique called

Model Tree Ensemble (MTE, Jung et al., 2011). The MTE was trained using information on meteorological conditions, remote-sensed information on vegetation intensity (fAPAR), and gridded information about vegetation type. Information on plant nitrogen status is not included in the training, therefore the GPP signal inferred from the MTE-GPP product does not explicitly reflect nitrogen-induced spatial patterns. Similarly, because MTE is trained once for all years from 1982 to 2008 and that neither atmospheric $[CO_2]$ nor nitrogen availability are predictors in the training, MTE-GPP has no temporal trend

which may be attributed indirectly or directly to these two driving variables.

### 2.3 Data used as model forcing variables

#### 2.3.1 Meteorological data

In-situ meteorology, which is typically observed at individual Fluxnet stations, was used to drive the ORCHIDEE simulations at the Fluxnet stations. In-situ data were gap-filled as described in Vuichard and Papale (2015) in order to

provide continuous half-hourly records of temperature, precipitation, short- and long-wave incoming radiation, wind speed and specific humidity to the ORCHIDEE model.

Global scale simulations were driven by CRU-NCEP meteorological data, available for the 1901-2016 period. CRU-NCEP consists of 6-hourly meteorological fields from the NCEP/NCAR reanalysis at 0.5 x 0.5 degree resolution that was bias

corrected with monthly CRU data.

#### 2.3.2 Vegetation-related information

For the site simulations at Fluxnet stations, the selection of the Plant Functional Type used to characterize vegetation at each site was based on in-situ information gathered within the Fluxnet dataset, using the International Geosphere-Biosphere Programme (IGBP) classification. For the global scale simulations, the PFT distribution within each grid cell over the time

period 1860-2016 was derived from History Database of the Global Environment (HYDE v3.2; Goldewijk et al., 2017) for the crop and pasture extend and from Olson et al. (1985) for the specification of the forest types.

#### 2.3.3 Soil data

The soil-related data used for driving ORCHIDEE are texture class, pH and bulk density at 0.5 x 0.5 degree resolution. Texture class was coming from Zobler (1986), bulk density from the Harmonized World Soil Database (HWSD,

FAO/IIASA/ISRIC/ISSCAS/JRC, 2012) and soil pH from International Geosphere-Biosphere Programme Data Information



System Soil Data (Global Soil Data Task Group, 2000). These different global datasets were used for both site-level and global simulations.

### 2.3.4 Nitrogen deposition data

Monthly atmospheric nitrogen deposition ($NH_x$ and $NO_y$) during 1860 - 2014 were taken from the IGAC/SPARC Chemistry-

Climate Model Initiative (CCMI, Eyring et al., 2013). Nitrogen deposition fields are available globally at a resolution of 0.5 x 0.5 degree. In the CCMI models, nitrogen emissions from natural biogenic sources, lightning, anthropogenic sources, biomass burning are accounted for, as well as the atmospheric transport of nitrogen gases. The CCMI product is used in the global $N_2O$ Model Intercomparison Project (NMIP, Tian et al., 2018) and is the official product for CMIP6 models without interactive chemistry components. ORCHIDEE reads total (wet and dry) nitrate ($NO_y$) and total ammonium ($NH_x$)

atmospheric deposition rates from the CCMI product and uses this information to drive the nitrogen cycle within the soil-plant continuum. In ORCHIDEE, nitrate and ammonium are added to the respective soil mineral nitrogen pools at each model's time step, omitting nitrogen interception by the canopy.

### 2.3.5 Nitrogen fertilizer data

Global simulations were also driven by nitrogen application under the form of synthetic fertilisation or manure. Synthetic

nitrogen fertilizer gridded annual data from 1960 to present-day were developed within the $N_2O$ Model Intercomparison Project (Lu and Tian, 2017) and is based on national-level data from the International Fertilizer industry Association (IFA) and the FAO. Nitrogen fertilizer application rate between 1860 and 1960 were extrapolated assuming that gridded values for 1960 were linearly reduced to the zero in 1900. For nitrogen fertilization through manure application, gridded annual data were compiled and downscaled based on country-level livestock population data from FAO (Zhang et al., 2017). Note that

the carbon and nitrogen contained in manure represents a lateral flux from grasslands to croplands. When closing the carbon and nitrogen budgets or calculating the net biome production – not applicable to this study, manure and synthetic nitrogen fertilizer should thus be accounted for in different ways.

### 2.4 Simulation set-up

### 2.4.1 Spin-up procedure

Each simulation requires a spin-up during which the model state variables (e.g., soil and biomass carbon and nitrogen pools) are put at equilibrium. Given that the time period needed to reach equilibrium by far exceeds the length of the available in-situ meteorological records, the spin-up at Fluxnet stations cycles several times over the entire record of in-situ meteorological observations. The spin-up started with a 500-year long run that makes use of the semi-analytical spin-up (see Sect. 2.1) in order to get the fluxes of litter fall but also the soil organic carbon pools at equilibrium, for $[CO_2]$ atmospheric

concentration and nitrogen deposition of the year 1860. Following this steady-state simulation, the spin-up continued with a





transient simulation from 1860 up to the start of the observation period for each site, varying $[CO_2]$ atmospheric concentration and the nitrogen deposition with historical data and still cycling the meteorological in-situ data.

Likewise, a semi-analytical spin-up for the global simulations was performed for the conditions of the year 1860, by using the nitrogen input data (deposition and fertiliser fields), the $[CO_2]$ value and land-use of the year 1860. Because CRU-NCEP is only available from 1901, data for the period 1901-1920 was used for the spin-up. As for site simulations, we performed a transient simulation varying the different fields driving the GPP evolution (see below).

### 2.4.2 Reference simulations

From the end of the transient phase at Fluxnet stations, a set of simulations (one at each Fluxnet station) was performed over
the observation period with $[CO_2]$ and nitrogen deposition level (from the CCMI monthly dataset) corresponding to this period. These simulations for present-day conditions (pd), in which carbon and nitrogen cycles are fully coupled (CNdyn configuration), are named pd-CNdyn.

Likewise, from the global steady state corresponding to the end of the spin-up simulation, a transient simulation from 1861
to 2016 is ran, accounting for climate change, $[CO_2]$ rise, land use change (LUC) and evolution of nitrogen atmospheric deposition (Ndep), nitrogen synthetic fertiliser (Nfert) and manure (Nman) on a yearly basis. CRU-NCEP data for the period 1901-1920 was used for simulating the period from 1861 to 1901, while from 1901 onwards the climatic forcing from the matching year was used. This simulation, which accounts for carbon-nitrogen interactions, is named S1-CNdyn.

### 2.4.3 Sensitivity test simulations

At each Fluxnet station, different simulations were performed in order to test the sensitivity of ORCHIDEE to different processes or forcing variables. As described in Eq. (3), leaf nitrogen content and leaf C/N ratio affect the carbon assimilated by photosynthesis, which in turn affects the leaf area index and, feeds back to carbon assimilation. Consequently, a set of two simulations was ran to investigate the impact of constraining the leaf C/N ratio on GPP, respectively in time and in time and space. One simulation is named pd-CNfix-time (based on the CNfix configuration) in which the leaf C/N ratio was fixed
to the time-average value at site level, which was in turn inferred from the pd-CNdyn reference simulation. In the other simulation named pd-CNfix-timePFT simulation (also based on the CNfix configuration), the leaf C/N ratio was set to the time-average PFT-average which was also inferred from the pd-CNdyn reference simulations.

The sensitivity of the simulated GPP to $[CO_2]$ increase (keeping the other drivers as constant) was tested by comparing four
idealized 100-year long simulations against control simulations for which atmospheric $[CO_2]$ was held constant at its present day concentration. A set of simulations started from present-day atmospheric $[CO_2]$ and $[CO_2]$ was increased by 1% per year (labelled 1%CO2). With present-day $[CO_2]$ level around 380 ppm, one percent increase per year leads almost to a tripling of





[$CO_2$] between the start and the end of the simulation. In the other set of simulations (labelled 2xCO2), [$CO_2$] was set to twice its present-day value along the 100 years. Both set-ups were repeated for the CNdyn and CNfix-time configurations, thus, resulting in a total of four idealized tests (1%CO2-CNdyn, 1%CO2-CNfix-time, 2xCO2-CNdyn and 2xCO2-CNfix-time simulations, see Table 2). For these sensitivity tests, climate drivers and nitrogen deposition files corresponding to the period of in-situ observations were used cyclic.

At global scale, in addition to the reference S1-CNdyn simulation, sensitivity tests were set-up to disentangle the main drivers of the simulated global increase in GPP (see Table 2). Based on the $N_2O$ Model Intercomparison Project protocol (Tian et al., 2018), additive scenarios were developed where GPP is only driven by climate change (S6-CNdyn); climate change and [$CO_2$] increase (S5-CNdyn); climate change, [$CO_2$] increase and land use change (S4-CNdyn); climate change, [$CO_2$] increase, land use change and nitrogen deposition evolution (S3-CNdyn); climate change, [$CO_2$] increase, land use change, nitrogen deposition and nitrogen fertilizer evolution (S2-CNdyn) using the different forcing data presented in Sect. 2.3. A control simulation named S0-CNdyn was also performed, which extends the spin-up simulation (i.e., using the nitrogen input data, the [$CO_2$] value and land-use of the year 1860 and recycling the meteorological data of the period 1901-1920).

To test the impact on the GPP evolution of accounting for the carbon-nitrogen interactions, an additional simulation was run at global scale in which the gridded leaf C/N ratio was fixed to the time-average value for each PFT, inferred from the S0-CNdyn control simulation. Thus, the plant nitrogen status of each PFT within each grid-cell is the one of the pre-industrial era (i.e. for the year 1860). In this simulation, all drivers were varying over the period 1861-2016. However, the GPP being un-sensitive to variation of nitrogen deposition, nitrogen fertilization, and N from manure in the CNfix configuration, this simulation corresponds to a S4 scenario and is consequently named S4-CNfix.

## 2.5 Evaluation metrics

The mean seasonal cycle of the simulated GPP was evaluated against observations. This was done at the PFT level by computing the mean seasonal cycle, averaged for all years and all sites belonging to a PFT. Alternatively, simulated daily GPP fluxes were evaluated by computing their root mean squared deviation (RMSD) to the observations. Further, the mean squared deviation (MSD) of the daily fluxes to the observation was decomposed and the contribution to the overall MSD from the mean bias, standard deviation and correlation was quantified based on the method of Kobayashi and Salam (2000). MSD can be written as:

$$MSD = SB + SDSD + LCS \,, \tag{13}$$

where SB is the squared bias, SDSD is the squared difference between standard deviations and LCS the lack of correlation weighted by the standard deviations. SB, SDSD and LCS reflect respectively errors on bias, standard deviation and



correlation. Finally, annual mean GPP flux was computed at each Fluxnet site in order to evaluate the model capacity at simulating site-to-site variations within each PFT.

The global S1-CNdyn simulation was also evaluated by comparing the spatial distribution of the annual mean GPP over the
period 2005-2016 to the one of the MTE-GPP product. Furthermore, time evolutions of the annual GPP, simulated by ORCHIDEE and derived from MTE-GPP were compared for different regions: globally, Northern hemisphere (>25°N), Southern hemisphere (<25°S) and tropical regions (<25°N and >25°S).

Lastly, the relative contributions of the different drivers to the present-day GPP value were assessed by computing the
successive differences between additive scenarios. Thus, the contributions of climate change, [$CO_2$] increase, land use change, nitrogen deposition, nitrogen fertilization and nitrogen manure were provided respectively by the differences between S6 and S0, S5 and S6, S4 and S5, S3 and S4, S2 and S3, S1 and S2. Using this methodology, the sum of the individual contributions is additive and equals the present-day GPP, thus ignoring possible non-linear interactions between drivers.

## 3 Results

### 3.1 Site-level simulations

#### 3.1.1 Evaluation of the standard configuration

The mean seasonal cycle of the GPP averaged per Plant Functional Type was reproduced for most PFTs, in the pd-CNdyn simulations (Fig. 1a). Over Tropical Evergreen Broadleaved Forests, the observed mean GPP values do not show a marked
seasonal cycle, while the simulated GPP are 3.0 gC m$^{-2}$ day$^{-1}$ higher in May-June compared to the rest of the year. For Temperate Evergreen Needleleaved Forests, the mean seasonal cycle simulated by ORCHIDEE matched the observations in terms of correlation (correlation coefficient of 0.99) and amplitude (model standard deviation of 2.4 gC m$^{-2}$ day$^{-1}$ to compare to 2.1 for observations), although the simulated GPP was overestimated by ~30% all the year. The model has the weakest performance for Temperate Evergreen Broadleaved Forest shown by a too pronounced seasonal cycle compared to that
observed (model standard deviation of 2.1 gC m$^{-2}$ day$^{-1}$ to compare to 0.7 gC m$^{-2}$ day$^{-1}$ for observations). In winter and early spring, GPP was overestimated by 2.0 gC m$^{-2}$ day$^{-1}$, while later in June and July the decrease of the GPP was overestimated. Simulated and observed mean seasonal cycles match each other for Temperate Deciduous Broadleaved Forests, the main discrepancy being a delay of ~10 days of the onset and senescence phases of the simulated GPP. The simulated mean seasonal cycle of the GPP over Boreal Evergreen Needleleaved Forests overestimated GPP by ~35% all the year while the
variations of the simulated GPP were in agreement with those observed (correlation coefficient of 0.96 and model std of 2.4 gC m$^{-2}$ day$^{-1}$ compared to 1.9 gC m$^{-2}$ day$^{-1}$ for observations). Last, over natural C3 grasslands, the increase of GPP at the





early stage of the growing season simulated by ORCHIDEE was too slow, compared to observations and the simulated GPP maintained its maximal value too long and too late in the season. This results in a low correlation between model and observation (correlation coefficient of 0.81)

The RMSE of the daily GPP flux averaged per PFT does not exceed 2.5 gC m$^{-2}$ day$^{-1}$ (Fig. 1b). The annual productivity of the PFTs being significantly different, the RMSE, when expressed as a percentage of the mean annual GPP (NRMSE), varies from 25% for Tropical Evergreen Broadleaved Forests to 80% for Boreal Evergreen Needleleaved Forests. Figure 1c shows the respective contributions of SB (bias), SDSD (deviation) and LCS (correlation) on the total MSE per PFT. These relative contributions to the MSE differed depending of the PFT. Error on the mean bias was the largest contribution to MSE for

Temperate and Boreal Evergreen Needleleaved Forests. At Tropical Evergreen Broadleaved Forests and C3 grassland sites, errors on the correlation contributed the most to the MSE while at Temperate Evergreen and Deciduous Broadleaved Forest sites, the three sources of error were more equally distributed.

The simulated annual mean GPP per site was comparable to the one observed for most sites (Fig. 1d), with a RMSE

averaged per PFT, which varied from 409 gC m$^{-2}$ yr$^{-1}$ (for Tropical Evergreen Broadleaved Forests) to 759 gC m$^{-2}$ yr$^{-1}$ (for Temperate Evergreen Broadleaved Forests). Both overestimation as well as underestimation were observed in all PFTs suggesting small systematic biases. Site-to-site variations of the annual mean GPP were relatively well reproduced for Tropical Evergreen Broadleaved Forests, Temperate Evergreen Needleleaved and Broadleaved Forests and C3 grassland sites sites (Pearson's correlation coefficient of 1.0 but for only two sites, 0.63, 0.44 and 0.82, respectively) but not for

Temperate Deciduous Broadleaved Forest and Boreal Evergreen Needleleaved Forest sites (Fig. 1d). For Temperate Deciduous Broadleaved Forest sites, the model produces significantly larger site-to-site GPP differences than the observations.

### 3.1.2 Sensitivity to model configurations

The leaf C/N ratio in the pd-CNdyn simulation varied substantially over time and/or from one site to another (Fig. 2c). The

observed variation was partly driven by the different nitrogen deposition load (Fig. 2a and b) as well as by differences in the simulated mineralisation and plant Nitrogen uptake (not shown). $V_{cmax}$ being directly related to the leaf nitrogen content, the variations of the leaf C/N ratio induced seasonal variations of the $V_{cmax}$ on the order of 0 to 30%, depending of the sites.

The mean and median values of the MSE of the simulated daily GPP obtained from the range of sites within a vegetation

class did not change substantially depending on whether nitrogen dynamics were accounted for or were fixed over time (pd-CNdyn, pd-CNfix-time or pd-CNfix-timePFT) (Fig. 3a). This finding holds for the error measures when decomposed into bias, standard deviation and correlation (Figs 3b-d). One exception to this model behaviour which is common across configurations is for some C3 grassland sites, where MSE and all its components were higher in the pd-CNfix-time and pd-



CNfix-timePFT simulations compared to the pd-CNdyn simulation (Fig. 3). For Tropical and Temperate Evergreen Broadleaved Forest classes, the CNfix-time simulation exhibits narrower ranges for MSE or for some of its components (SB, SDSD, or LCS) compared to the pd-CNdyn simulation. This highlights the fact that, for some of the Tropical and Temperate Evergreen Broadleaved Forest sites, constraining the leaf C/N ratio in time improved the fit to the observed GPP, while the
same constraint deteriorated the fit at some other sites within that PFT. Results obtained with the pd-CNfix-timePFT simulation also lead to slightly narrower ranges of MSE values and of its bias (SB) subcomponent for Temperate Evergreen Needleleaved Forest, compared to the pd-CNfix-time simulation.

Simulated site-to-site variations of the annual mean GPP were sensitive to the model configuration regarding the leaf C/N
ratio (Fig. 4). For all PFTs except the Boreal Evergreen Needleleaved Forest sites, the pd-CNdyn is the simulation, which best matched the observations in terms of RMSE. Nevertheless, the RMSE were of the same order of magnitude for the three model configurations for all PFTs except for C3 grassland sites for which the RMSE was significantly lower in the pd-CNdyn simulation (Fig. 4).

### 3.1.3 Sensitivity to [$CO_2$] increase

Increasing atmospheric [$CO_2$] by 1% per year lead to a continuous increase of the GPP in any of the two configurations; one with a dynamic leaf C/N ratios (1%CO2-CNdyn) and the other with leaf CN ratios fixed over time (1%CO2-CNfix-time) and for any PFT class (Fig. 5). Note first that the large temporal cycle for the Tropical Evergreen Broadleaved forest is due to the recycling of two to four years of in situ meteorological forcing. As expected, the higher the [$CO_2$], the higher the GPP was. Nevertheless, the GPP increase was less sensitive to the [$CO_2$] increase in the configuration with a dynamic C/N ratio
(1%CO2-CNdyn), which reflects an induced Nitrogen-limitation of the photosynthesis. Notably, the GPP sensitivity to $CO_2$ increase in the 1%CO2-CNdyn simulation was particularly low for Boreal Evergreen Needleleaved Forest class. After 100 years of [$CO_2$] increase, the difference in GPP increase between the 1%CO2-CNfix-time and 1%CO2-CNdyn simulations reached 1.0 kgC m$^{-2}$ yr$^{-1}$ for Tropical Evergreen Broadleaved Forests 0.8 kgC m$^{-2}$ yr$^{-1}$ for Temperate Evergreen Needleleaved Forest, 0.8 kgC m$^{-2}$ yr$^{-1}$ for Temperate Evergreen Broadleaved Forest, 0.6 kgC m$^{-2}$ yr$^{-1}$ for Temperate Deciduous Broadleaved
Forest, 0.6 kgC m$^{-2}$ yr$^{-1}$ for Boreal Evergreen Needleleaved Forest and 1.0 kgC m$^{-2}$ yr$^{-1}$ for C3 natural grasslands. These differences in GPP increase corresponded to values of the order of 30-50% of the present-day annual mean GPP.

In the simulation where [$CO_2$] was doubled compared to the present-day level but N-limitation was not accounted for (2xCO2-CNfix-time), GPP increased by an overall 90% (1.1 kgC m$^{-2}$ yr$^{-1}$, Fig. 5), and at some sites even by as much as
150%. Accounting for a N-limitation (2xCO2-CNdyn simulation) reduced the overall GPP increase to ~50% compared to present-day value (0.6 kgC m$^{-2}$ yr$^{-1}$, Fig. 5). A decreasing trend in GPP increase for all PFTs - except Temperate Deciduous Broadleaved Forest and C3 grassland sites - was apparent for the 2xCO2-CNdyn simulation, while such a trend was absent in the simulations without carbon-nitrogen interactions (2xCO2-CNfix-time). For instance, mean GPP increase at Temperate





Evergreen Needleleaved Forest sites reached 0.8 kgC m$^{-2}$ yr$^{-1}$ over the 5 years consecutive to the doubling of the CO2 level but was only 0.5 kgC m$^{-2}$ yr$^{-1}$ after 100 years (Fig. 5). Furthermore, when comparing the simulations without and with N-limitations, the year-to-year variability in GPP was amplified compared to the present-day variability for the configuration without N-limitation.

GPP increase induced by increase of atmospheric [CO$_2$] was achieved at a limited water cost or even resulted in saving water, at most sites. In the 1%CO2-CNfix-time simulations, after 100 years of [CO$_2$] increase, transpiration rate averaged per PFT was lower by few millimetres up to 50 mm per year, compared to its value in pd-CNfix-time simulation (Fig. 6, left column). Averaged over all PFT, the mean transpiration rate decreased by ~25 mm yr-1 after 100 years. In the 1%CO2-

CNdyn simulation, the decrease of the transpiration rate was even stronger (Fig. 6). Mean transpiration rate decrease averaged per PFT reached 75 mm to 110 mm per year, after 100 years. Averaged over all sites, the mean decrease equalled 100 mm per year, which corresponds to ~20% of its value in the pd-CNdyn simulation. Similar model behaviour with the 2xCO2 type simulations was exhibited (Fig. 6, right column). Compared to its value in the pd-CNfix-time simulation, the mean transpiration rate averaged over all sites was stable in the 2xCO2-CNfix-time simulation. In the 2xCO2-CNdyn

simulation, the mean transpiration rate was 50 mm per year lower than it was in the pd-CNdyn simulation (15%).

When expressed in terms of water use efficiency (WUE, unitless) - defined here as the ratio of GPP (gC m$^{-2}$ yr$^{-1}$) to transpiration (gH$_2$O m$^{-2}$ yr$^{-1}$) – these model responses translated into an increase of the WUE for all the sensitivity tests (Fig. 7). Under the 1%CO2-CNfix-time simulation, after 100 years, WUE increased by 120% on average per PFT (Fig. 7, mean

increase of 120%) compared to the pd-CNfix-time simulation. After 100 years, the mean WUE increase in the 1%CO2-CNdyn simulation is 17% lower than in the 1%CO2-CNfix-time simulation. Similarly, the mean WUE averaged per PFT in the 2xCO2-CNfix-time simulation is 70% to 90% higher than in the pd-CNfix-time simulation (mean increase of 80% over all sites, Fig. 7). In the 2xCO2-CNdyn simulation, the mean increase of the WUE is 10% lower than in the 2xCO2-CNfix-time simulation.

**3.2 Global scale simulations**

The present-day annual mean GPP under the S1-CNdyn simulation reached its maximum values in Central Africa (Fig. 8a). Compared to values reported by MTE-GPP (Jung et al. 2010), simulated annual mean GPP in this region is overestimated by up to 2.0 kgC m$^{-2}$ yr$^{-1}$ (Fig. 8b). On the other hand, annual mean GPP values in the Amazon region or Indonesia were underestimated by 1.5 kgC m$^{-2}$ yr$^{-1}$. Overall, the Root Mean Square Deviation (RMDS) to the MTE-GPP values equals 0.7

kgC m$^{-2}$ yr$^{-1}$, compared to the global mean annual flux from MTE-GPP of 1.0 kgC m$^{-2}$ yr$^{-1}$. With the difference between the simulations and MTE-GPP data being less than 0.5 kgC m$^{-2}$ yr$^{-1}$ in most extra-tropical regions, the Root Mean Square Deviation is thus dominated by the mismatch between the simulations and observations over the tropics.





When averaged by latitudinal band, the overlap between simulated annual mean GPP and MTE-GPP estimates increases (Fig. 9) compared to the overall global mapping result. From 1982 to 2008, the simulated annual mean GPP above 25°N increased from 36 PgC yr$^{-1}$ to 42 PgC yr$^{-1}$ (around 17% increase), with large year-to-year variations (up to 3 PgC yr$^{-1}$) (Fig. 9a). Over the same period and the same domain, the MTE-GPP estimates varied between 37 and 40 PgC yr$^{-1}$ (around 8%

increase) thus showing a weaker positive trend compared to the ORCHIDEE simulation. In the tropical regions (between 25°S and 25°N), the MTE-GPP estimate slightly increased from 72 PgC yr$^{-1}$ to 75 PgC yr$^{-1}$ (around 4% increase) while according to the ORCHIDEE S1-CNdyn simulation, GPP between 1982 and 2008 varied from 62 PgC yr$^{-1}$ to 72 PgC yr$^{-1}$ (around 16% increase) (Fig. 9b). In the tropical regions, the ORCHIDEE GPP strongly increases between 1992 and 2000 (from 64 to 72 PgC yr$^{-1}$) and then fluctuates between 69 and 72 PgC yr$^{-1}$ after 2000.

Due to a lower land area and a lower productivity per unit of land, the contribution of the southern lands (below 25°S) to the global GPP is rather small. It amounts 5 PgC yr$^{-1}$ with ORCHIDEE, and 6 PgC yr$^{-1}$ with MTE-GPP over the 1982-2008 period (Fig. 9c). Both estimates have no trend over the time period and the year-to-year variations are slightly larger with the ORCHIDEE model (standard deviation of 0.3 PgC yr$^{-1}$ compared to 0.2 PgC yr$^{-1}$ with MTE-GPP). Globally, annual mean

GPP reached ~120 PgC yr$^{-1}$ in 2016 (Fig. 9d).

Over the period 1861-2016, when accounting for all driving variables (S1-CNdyn simulation), simulated global GPP increased by ~50%, from 80 PgC yr$^{-1}$ to 120 PgC yr$^{-1}$ (Fig. 10a, 39% over the 20$^{th}$ century). When only driven by climate change and [$CO_2$] increase (S5-CNdyn simulation), simulated GPP varied from 80 PgC yr$^{-1}$ in 1861 to 104 PgC yr$^{-1}$, in 2016

(~30% increase). Similarly, without the increase in nitrogen fertilisation (S3-CNdyn simulation), global GPP would only have increased by 34% over the same period. The relative contributions of the different drivers to the GPP growth over 1861-2016 were 10% for climate change, 50% for [$CO_2$] increase, -13% for land-use change, 20% for nitrogen deposition change, and 33% for nitrogen fertilisation change. Global GPP evolution under the S4-CNfix simulation is similar to that under the S1-CNdyn simulation (Fig. 10a). These simulated GPP responses to variations of driving factors are not equally

distributed across PFTs. Thus, simulated GPPs over all forested lands (Fig. 10b) and over all grasslands and croplands (Fig. 10c) show contrasted responses. In particular, simulated present-day GPP over forested lands equals ~63 PgC yr$^{-1}$ under the S1-CNdyn simulation (Fig. 10b) and ~82 PgC yr$^{-1}$ under the S4-CNfix simulation. On the opposite, GPP over grasslands and croplands (Fig. 10c) equals 56 PgC yr$^{-1}$ under the S1-CNdyn simulation, but only 43 PgC yr$^{-1}$ under the S4-CNfix simulation.

**4 Discussion**

We presented revision 4999 of the ORCHIDEE model that accounts for carbon-nitrogen interactions based on the initial work of Zaehle and Friend (2010). In this model version, improvements were implemented with respect to water, energy and





carbon budgets as well as to the nitrogen cycle and its coupling to the carbon cycle, compared to the original version of ORCHIDEE used in Zaehle and Friend (2010). Among these changes, a new carbon assimilation scheme was introduced (Yin and Struik, 2009) in which the maximum Rubisco activity-limited carboxylation rate is a direct function of the leaf nitrogen content (Kattge et al., 2009). Because these model developments primary impact on the GPP and because this flux

is the primary carbon input flow into the ecosystem, we focused our study on an in-depth evaluation of the simulated GPP flux, from site to global scale, and from daily to mean annual time scale.

We showed that the version of ORCHIDEE (r4999) that accounts for carbon-nitrogen interactions can simulate reasonably well the mean seasonal cycle, daily mean variations, and annual mean GPP for most PFTs. The overlap between the

ORCHIDEE simulations and the eddy-covariance based observations is similar to other models (Balzarolo et al., 2014; Slevin et al., 2015). For instance, based on an evaluation over 32 Fluxnet sites, Balzarolo et al. (2014) reported mean RMSE values of the daily GPP flux simulated by three GTEMs (ie. ORCHIDEE, CTESSEL and ISBA-A-gs) which range between 1.9 and 4.4 gC m$^{-2}$ d$^{-1}$ depending of the model and PFT considered (excluding cropland sites for which all models performed the least and which are not included in our study). In comparison, mean RMSE of the simulated daily GPP flux in our study

for any PFT never exceeded 2.5 gC m$^{-2}$ d$^{-1}$.

The exception to the general agreement is for the Temperate Evergreen Broadleaved Forest sites for which the ORCHIDEE r4999 failed to reproduce the observed mean seasonal cycle. Indeed, the weak performances are for the five Mediterranean sites, but not for the two Australian sites. The Mediterranean sites receive high N-deposition loads leading to low leaf C/N

ratios. Over these sites, at the beginning of the growing season, the higher maximum Rubisco activity-limited carboxylation rate values ($Vc_{max}$) induced by high leaf nitrogen content is the main reason explaining the GPP overestimation. The overestimation of the GPP at the early stage of the growing season induced higher rates of transpiration. These higher rates of transpiration partly explain the underestimation of the GPP during the summer, due to a too strong depletion of the soil water. Because the too low GPP during the summer demands less nitrogen to support biomass growth, this results in higher

nitrogen available later in the season. This mechanism tends to maintain or even amplify the mismatch at the beginning of the growing season. The two Australian sites receive low N-deposition loads in comparison to the three Mediterranean sites. Consequently, the Australian sites have high leaf C/N ratios (see Fig. 2c, for TeEBF where Australian sites correspond to the blue and green lines), which overcomes the overestimate of the GPP at the beginning of the growing season and the subsequent issues.

Global scale annual mean GPP simulated by ORCHIDEE and aggregated per latitudinal band, matches GPP estimated by the MTE-GPP product. The temporal increase in GPP projected by ORCHIDEE for the northern and tropical regions are not apparent from the MTE-GPP. Indeed, the MTE-GPP is only driven by the climate variability, vegetation greenness index and land cover information. As such, MTE-GPP does not represent directly the effect of $CO_2$ and nitrogen on the GPP but only



indirectly through changes in vegetation greenness. This conceptual limitation may explain the absence of trends in the GPP signal from MTE-GPP and restricts the domain of validity of this product to the period over which it has been trained. The aforementioned limitation may also explain part of the spatial disagreement between ORCHIDEE and MTE-GPP.

Apart from the MTE-GPP product, there is few spatially explicit data available that is suitable to evaluate global simulations of GPP. Due to the fact that two $CO_2$ fluxes occur at land but have an opposite direction (ie GPP and Total Ecosystem Respiration) data of atmospheric $[CO_2]$ cannot be used for this purpose. As an alternative, Campbell et al. (2017) assessed the GPP growth rate over the 20[th] century based on an independent estimate by using atmospheric carbonyl sulfide (COS) records. Based on a procedure that optimizes the COS sources and sinks to match the observed atmospheric [COS] dynamic,

they deduce that the GPP increased by 31%±5% over the 20[th] century. This corresponds to a larger GPP growth rate than that estimated by most Global Terrestrial Ecosystem Models. Although higher than the 31%±5% based on the COS constraint, the ORCHIDEE based of 39% is in good agreement with the COS constraint compared to most of other ecosystem models reported in the study of Campbell et al. (2017). This increases our confidence in the ability of ORCHIDEE r4999 to account for the interactions between carbon and nitrogen.

Constraining the leaf C/N ratio over time and per PFT (using the values from a simulation with dynamic C/N ratios) was found not to impact model performance at site-level, for most PFTs. Note, however, that the leaf C/N ratios in the pd-CNfix-time and pd-CNfix-timePFT simulations were set using the present-day values from the pd-CNdyn simulation. At some sites, the mean standard error of the simulated daily mean GPP variation is reduced, but an increase in mean standard error was

observed at other sites. At first, the similarity in performance for configurations with and without dynamic leaf C/N ratio may appear as a failure in term of expected impacts with the inclusion of the nitrogen cycle in the model. Accounting for dynamic leaf C/N ratio and for site-specific information such as nitrogen deposition could be expected, at least from a theoretical point of view, to enhance the predictive power of ORCHIDEE. However, at the same time, dynamic modelling of the leaf C/N ratio adds complexity and interactions, which come with additional sources of uncertainties. Our results indicate

that for some sites the increase in predictive power dominates whereas for other sites the additional uncertainties dominate the total model error.

One exception were the C3 grassland sites for which the mean model performance was lower when carbon-nitrogen interactions were not accounted for (pd-CNfix-time and pd-CNfix-timePFT simulations), leading to a higher mean MSD for

this PFT compared to the pd-CNdyn simulation. Constraining or not the leaf C/N ratio (pd-CNfix-time vs. pd-CNdyn configuration) alters GPP via changes on the $Vc_{max}$ value. Moreover, for some "extreme" conditions in the CNdyn configuration, the carbon allocation to the different reservoirs may be reduced due to insufficient nitrogen in the labile pool, irrespective of the value of the C/N ratio of the standing leaf biomass (see end of Sect. 2.1.3 where this model behaviour is detailed). These extreme conditions and their consequences on the biomass allocation cannot be captured with the CN-fix



configuration and it appears that at many C3 grassland sites, these extreme conditions are sufficiently frequent to produce different model responses and performances between the pd-CNfix-time and pd-CNfix-timePFT simulations and the pd-CNdyn simulation.

While behaviour and performance are similar across ORCHIDEE configurations for present-day conditions for most PFTs, they differ substantially in terms of the response of GPP to enhance atmospheric [$CO_2$]. Site-mean increase in GPP after 100 years under the 1%CO2 experiment is half when carbon-nitrogen interactions are accounted for compared to ignoring the carbon-nitrogen interactions (CNdyn, compared to CNfix-time). Interestingly, this effect-size is of the same order of magnitude as the simulated reduction in the increase in NPP (65%) under the Representative Concentration Pathway 8.5
(Wieder et al., 2015) estimated by coupled carbon–climate model projections and C/N stoichiometric models. The reduction in the increase in NPP was caused by the interaction between enhanced atmospheric [$CO_2$] and nitrogen limitations trends (Wieder et al., 2015).

Simulated site-average GPP increased by 50% when [$CO_2$] was doubled (~+370 ppm, 2xCO2 experiment) and carbon-
nitrogen interactions were accounted for. A similar response has been reported for some experimental FACE sites (Norby et al., 2010; Zaehle et al., 2014) where GPP increased by 30% for an increase between 150 and 200 ppm in atmospheric [$CO_2$]. Also, the simulated decrease of the growth rate in GPP over time at some sites is in line with the Progressive Nitrogen Limitation as postulated by (Luo et al., 2004). At global scale, we showed that the present-day simulated GPP is reduced by 20% when the additional nitrogen limitation associated to the period 1860-2016 is accounted for (S4-CNdyn) compared to
ignoring this nitrogen limitation (S4-CNfix). We showed that the simulated GPP trajectories from 1861 to 2016 under the S1-CNdyn and the S4-CNfix (where the leaf C/N ratios are fixed to their mean-time pre-industrial values) are similar, meaning that the global historical increase of nitrogen inputs (through deposition and fertilisation) was of the same order than the nitrogen demand to fulfil the GPP increase due to the [$CO_2$]-fertilisation effect. This general behaviour, however, hides contrasted responses between classes of PFTs. Over forested lands, the potential GPP increase primary driven by the
[$CO_2$]-fertilisation effect has been significantly limited due to a shortage in plant available nitrogen (compare S4-CNfix and S3-CNdyn curves of fig. 10b). On the opposite, over grasslands and croplands, the supply of nitrogen through fertilisation has fostered the GPP more than it would have done solely from the [$CO_2$]-fertilisation effect without N-limitation (compare S4-CNfix and S1-CNdyn curves of fig. 10c).

We showed that accounting for carbon-nitrogen interactions or not, when atmospheric [CO2] is enriched does not only impact on the GPP response but has also important consequences on the transpiration rate. We showed that on average, when atmospheric [$CO_2$] is doubled (~ 700-750 ppm), the water use efficiency - calculated as the ratio between GPP and transpiration - increases by 80% under the CNfix-time configuration, but only by 67% under the CNdyn configuration. The difference in water use efficiency change is likely the outcome of partly the nitrogen cycle through its impact on $Vc_{max}$ and





thus on stomatal conductance. Although based on a slightly different indicator, this is well in agreement with the observed increase (+68%) of the instantaneous transpiration efficiency (computed as the assimilation divided by the stomatal conductance, ITE) reported in the meta-analysis of Ainsworth and Long (2005) based on a set of C3 ecosystem sites where atmospheric [CO2] was enriched to reach atmospheric concentrations between 475 and 600 ppm. Our model experiments

illustrate the interplay between carbon, nitrogen and water cycles and highlight the need for careful analysis when modelling ecosystem productivity under future climate changes and its possible reduction due to water limitation (Ahlström et al., 2012).

## 5 Concluding remarks

We conclude that if carbon-nitrogen interactions are accounted for, the functional responses of ORCHIDEE r4999 better

agrees with current understanding of photosynthesis than when the carbon-nitrogen interactions are not accounted for. From this point of view our factorial experiment and sensitivity analysis confirm what has been shown by several other Global Terrestrial Models, i.e., that carbon-nitrogen interactions are essential in understanding global terrestrial ecosystem productivity (Goll et al., 2017; Sokolov et al., 2008; Thornton et al., 2009; Wang and Houlton, 2009; Wania et al., 2012; Wieder et al., 2015; Zaehle et al., 2010a, 2010b) and especially its dynamic under atmospheric [CO$_2$] increase and climate

change. Further simulations, making use of this new ORCHIDEE version and driven by different socio-economic scenarios over the 21$^{st}$ century (i.e. Representative Concentration Pathways) will be performed in order to better quantify the future GPP response to the combined evolution of [CO$_2$] and nitrogen land supply. Additionally, climate simulations with the IPSL Earth System Model (as part of the CMIP6 inter-comparison project) including this new ORCHIDEE version are planed to assess the impact of nitrogen limitation on the fate of the net land carbon sink and on climate projections.

**Code availability**

The source code is freely available online via the following address: http://forge.ipsl.jussieu.fr/orchidee/wiki/GroupActivities/CodeAvalaibilityPublication/ORCHIDEE_gmd-2018-261. A DOI has been requested for this page which provides guidelines for downloading.

**Appendix**

**Table A1: List of Fluxnet LaThuile sites used in this study and associated information related to country, location and corresponding Plant Functional Type into ORCHIDEE model**

| Site Id | Country | Site name | Latitude | Longitude | PFT |
|---|---|---|---|---|---|
| BR-Sa3 | Brazil | Santarem-Km83-Logged Forest | -3.018 | -54.971 | TrEBF |
| ID-Pag | Indonesia | Palangkaraya (PDF) | 2.345 | 114.036 | TrEBF |
| CZ-BK1 | Czech Republic | Bily Kriz- Beskidy Mountains | 49.503 | 18.538 | TeENF |



| | | | | | |
|---|---|---|---|---|---|
| DE-Bay | Germany | Bayreuth-Waldstein/WeidenBrunnen | 50.142 | 11.867 | TeENF |
| DE-Tha | Germany | Anchor Station Tharandt - old spruce | 50.964 | 13.567 | TeENF |
| DE-Wet | Germany | Wetzstein | 50.454 | 11.458 | TeENF |
| ES-ES1 | Spain | El Saler | 39.346 | -0.319 | TeENF |
| FR-LBr | France | Le Bray (after 6/28/1998) | 44.717 | -0.769 | TeENF |
| IL-Yat | Israel | Yatir | 31.345 | 35.052 | TeENF |
| IT-Lav | Italy | Lavarone (after 3/2002) | 45.955 | 11.281 | TeENF |
| IT-Ren | Italy | Renon/Ritten (Bolzano) | 46.588 | 11.435 | TeENF |
| IT-SRo | Italy | San Rossore | 43.728 | 10.284 | TeENF |
| NL-Loo | Netherlands | Loobos | 52.168 | 5.744 | TeENF |
| RU-Fyo | Russia | Fyodorovskoye wet spruce stand | 56.462 | 32.924 | TeENF |
| SE-Nor | Sweden | Norunda | 60.087 | 17.480 | TeENF |
| SE-Sk1 | Sweden | Skyttorp1 young | 60.125 | 17.918 | TeENF |
| SE-Sk2 | Sweden | Skyttorp | 60.130 | 17.840 | TeENF |
| SK-Tat | Slovak Republic | Tatra | 49.121 | 20.164 | TeENF |
| UK-Gri | UK | Griffin- Aberfeldy-Scotland | 56.607 | -3.798 | TeENF |
| US-Blo | USA | CA - Blodgett Forest | 38.895 | -120.633 | TeENF |
| US-Ho1 | USA | ME - Howland Forest (main tower) | 45.204 | -68.740 | TeENF |
| US-Ho2 | USA | ME - Howland Forest (west tower) | 45.209 | -68.747 | TeENF |
| US-Me4 | USA | OR - Metolius-old aged ponderosa pine | 44.499 | -121.622 | TeENF |
| US-SP2 | USA | FL - Slashpine-Mize-clearcut-3yr.regen | 29.765 | -82.245 | TeENF |
| US-SP3 | USA | FL - Slashpine-Donaldson-mid-rot- 12yrs | 29.755 | -82.163 | TeENF |
| US-SP4 | USA | FL - Slashpine-Rayonier-mid-rot- 12yrs | 29.803 | -82.203 | TeENF |
| US-Wi0 | USA | WI - Young red pine (YRP) | 46.619 | -91.081 | TeENF |
| US-Wi2 | USA | WI - Intermediate red pine (IRP) | 46.687 | -91.153 | TeENF |
| US-Wi4 | USA | WI - Mature red pine (MRP) | 46.739 | -91.166 | TeENF |
| US-Wi5 | USA | WI - Mixed young jack pine (MYJP) | 46.653 | -91.086 | TeENF |
| US-Wi9 | USA | WI - Young Jack pine (YJP) | 46.619 | -91.081 | TeENF |
| AU-Tum | Australia | Tumbarumba | -35.656 | 148.152 | TeEBF |
| AU-Wac | Australia | Wallaby Creek | -37.429 | 145.187 | TeEBF |
| FR-Pue | France | Puechabon | 43.741 | 3.596 | TeEBF |
| IT-Cpz | Italy | Castelporziano | 41.705 | 12.376 | TeEBF |
| IT-Lec | Italy | Lecceto | 43.305 | 11.271 | TeEBF |
| PT-Esp | Portugal | Espirra | 38.639 | -8.602 | TeEBF |
| PT-Mi1 | Portugal | Mitra (Evora) | 38.541 | -8.000 | TeEBF |
| DE-Hai | Germany | Hainich | 51.079 | 10.452 | TeDBF |
| DK-Sor | Denmark | Soroe- LilleBogeskov | 55.487 | 11.646 | TeDBF |
| FR-Fon | France | Fontainebleau | 48.476 | 2.780 | TeDBF |
| FR-Hes | France | Hesse Forest- Sarrebourg | 48.674 | 7.065 | TeDBF |
| IS-Gun | Iceland | Gunnarsholt | 63.833 | -20.217 | TeDBF |
| IT-Col | Italy | Collelongo- Selva Piana | 41.849 | 13.588 | TeDBF |
| IT-Non | Italy | Nonantola | 44.690 | 11.089 | TeDBF |
| IT-PT1 | Italy | Zerbolo-Parco Ticino- Canarazzo | 45.201 | 9.061 | TeDBF |
| IT-Ro1 | Italy | Roccarespampani 1 | 42.408 | 11.930 | TeDBF |
| IT-Ro2 | Italy | Roccarespampani 2 | 42.390 | 11.921 | TeDBF |
| UK-Ham | UK | Hampshire | 51.154 | -0.858 | TeDBF |
| UK-PL3 | UK | Pang/ Lambourne (forest) | 51.450 | -1.267 | TeDBF |
| US-Bar | USA | NH - Bartlett Experimental Forest | 44.065 | -71.288 | TeDBF |
| US-Ha1 | USA | MA - Harvard Forest EMS Tower (HFR1) | 42.538 | -72.172 | TeDBF |
| US-MMS | USA | IN - Morgan Monroe State Forest | 39.323 | -86.413 | TeDBF |
| US-MOz | USA | MO - Missouri Ozark Site | 38.744 | -92.200 | TeDBF |
| US-Oho | USA | OH - Oak Openings | 41.555 | -83.844 | TeDBF |
| US-UMB | USA | MI - Univ. of Mich. Biological Station | 45.560 | -84.714 | TeDBF |
| US-WBW | USA | TN - Walker Branch Watershed | 35.959 | -84.287 | TeDBF |
| US-WCr | USA | WI - Willow Creek | 45.806 | -90.080 | TeDBF |





| US-Wi8 | USA | WI - Young hardwood clearcut (YHW) | 46.722 | -91.252 | TeDBF |
| CA-Man | Canada | BOREAS NSA - Old Black Spruce | 55.880 | -98.481 | BoENF |
| CA-NS2 | Canada | UCI-1930 burn site | 55.906 | -98.525 | BoENF |
| CA-Qcu | Canada | Quebec Boreal Cutover Site | 49.267 | -74.037 | BoENF |
| CA-Qfo | Canada | Quebec Mature Boreal Forest Site | 49.693 | -74.342 | BoENF |
| CA-SF1 | Canada | Sask.- Fire 1977 | 54.485 | -105.818 | BoENF |
| FI-Hyy | Finland | Hyytiala | 61.847 | 24.295 | BoENF |
| FI-Sod | Finland | Sodankyla | 67.362 | 26.638 | BoENF |
| SE-Fla | Sweden | Flakaliden | 64.113 | 19.457 | BoENF |
| ES-VDA | Spain | Vall d'Alinya | 42.152 | 1.449 | GRA |
| FR-Lq2 | France | Laqueuille extensive | 45.639 | 2.737 | GRA |
| HU-Bug | Hungary | Bugacpuszta | 46.691 | 19.601 | GRA |
| IT-Amp | Italy | Amplero | 41.904 | 13.605 | GRA |
| IT-MBo | Italy | Monte Bondone | 46.016 | 11.047 | GRA |
| NL-Ca1 | Netherlands | Cabauw | 51.971 | 4.927 | GRA |
| NL-Haa | Netherlands | Haastrecht | 52.003 | 4.806 | GRA |
| NL-Hor | Netherlands | Horstermeer | 52.029 | 5.068 | GRA |
| RU-Ha1 | Russia | Ubs Nur- Hakasija-grassland | 54.725 | 90.002 | GRA |
| US-FPe | USA | MT - Fort Peck | 48.308 | -105.102 | GRA |
| US-Goo | USA | MS - Goodwin Creek | 34.255 | -89.874 | GRA |

**Author contribution**

NV and PM developed the model code with contributions from SL, SZ, BG and JG. NV performed the simulations. NV and VB prepared the figures. NV prepared the manuscript with contributions from all co-authors.

**Acknowledgements**

The work was funded by the EU FP7 project ERACLIM-2 and the EU H2020 project CRESCENDO. NV acknowledges Sophie Szopa for fruitful discussion and support. This work benefited from the HPC resources made available by GENCI (Grand Equipement National de Calcul Intensif), CEA. This work used eddy covariance data acquired by the FLUXNET community.

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





**Tables**

**Table 1. List of Plant Functional Types (PFT) used in the ORCHIDEE model and associated parameter values of Nitrogen Use Efficiency (NUE) and minimal and maximal leaf C/N ratio.**

| PFT acronym | PFT name | NUE ($\mu$mol $CO_2$ s$^{-1}$ [g$_{Nleaf}$]$^{-1}$) | $CN_{leaf,min}$ (g$_C$ [g$_N$]$^{-1}$) | $CN_{leaf,max}$ (g$_C$ [g$_N$]$^{-1}$) |
|---|---|---|---|---|
| TrEBF | Tropical Evergreen Broadleaved Forest | 14 | 16 | 45.5 |
| TrDBF | Tropical Deciduous Broadleaved Forest | 30 | 16 | 45.5 |
| TeENF | Temperate Evergreen Needleleaved Forest | 20 | 28 | 74.8 |
| TeEBF | Temperate Evergreen Broadleaved Forest | 33 | 16 | 45.5 |
| TeDBF | Temperate Deciduous Broadleaved Forest | 38 | 16 | 45.5 |
| BoENF | Boreal Evergreen Needleleaved Forest | 15 | 28 | 74.8 |
| BoDBF | Boreal Deciduous Broadleaved Forest | 38 | 16 | 45.5 |
| BoDNF | Boreal deciduous Needleleaved Forest | 22 | 16 | 45.5 |
| GraC3 | C3 grass | 45 | 16 | 45.5 |
| GraC4 | C4 grass | 45 | 16 | 45.5 |
| CroC3 | C3 crop | 60 | 16 | 45.5 |
| CroC4 | C4 crop | 60 | 16 | 45.5 |





**Table 2. List of simulations performed in this study and information related to time period, C/N configuration, and datasets used for climate, CO₂, land-use, nitrogen deposition, nitrogen synthetic fertiliser and nitrogen manure fertiliser.**

| Simulation Name | Domain | Period | CN configuration | Climate | CO₂ | Land Use | Nitrogen deposition | Nitrogen synthetic fertiliser | Nitrogen manure fertiliser |
|---|---|---|---|---|---|---|---|---|---|
| pd-CNdyn | In-situ | Observation period (from 1 to 15 yrs) | Dynamic | In-situ data | Fixed – Related to obs. period | Fixed – based on IGBP classification | Related to obs. period | / | / |
| pd-CNfix-time | In-situ | Observation period (from 1 to 15 yrs) | Fixed – Mean-time value from the pd-CNdyn simulation | In-situ data | Fixed – Related to obs. period | Fixed – based on IGBP classification | / | / | / |
| pd-CNfix-timePFT | In-situ | Observation period (from 1 to 15 yrs) | Fixed– Mean-time mean-PFT value from the pd-CNdyn simulation | In-situ data | Fixed – Related to obs. period | Fixed – based on IGBP classification | / | / | / |
| 1%CO2-CNdyn | In-situ | 100-year long simulation from obs. period | Dynamic | Recycling in-situ data | 1% yearly increase from obs. period CO2 | Fixed – based on IGBP classification | Recycling obs. period | / | / |
| 1%CO2-CNfix-time | In-situ | 100-year long simulation from obs. period | Fixed – Mean-time value from the pd-CNdyn simulation | Recycling in-situ data | 1% yearly increase from obs. period CO2 | Fixed – based on IGBP classification | / | / | / |
| 2xCO2-CNdyn | In-situ | 100-year long simulation from obs. period | Dynamic | Recycling in-situ data | Doubling of the obs. period CO2 | Fixed – based on IGBP classification | Recycling obs. period | / | / |
| 2xCO2-CNfix-time | In-situ | 100-year long simulation from observation period | Fixed – Mean-time value from the pd-CNdyn simulation | Recycling in-situ data | Doubling of the obs. period CO2 | Fixed – based on IGBP classification | / | / | / |
| S1-CNdyn | Global | 1861-2016 | Dynamic | CRU-NCEP 1901-2016 | 1861-2016 | 1861-2016 | 1861-2016 | 1861-2016 | 1861-2016 |
| S2-CNdyn | Global | 1861-2016 | Dynamic | CRU-NCEP 1901-2016 | 1861-2016 | 1861-2016 | 1861-2016 | 1861-2016 | 1861 |
| S3-CNdyn | Global | 1861-2016 | Dynamic | CRU-NCEP 1901-2016 | 1861-2016 | 1861-2016 | 1861-2016 | 1861 | 1861 |
| S4-CNdyn | Global | 1861-2016 | Dynamic | CRU-NCEP 1901-2016 | 1861-2016 | 1861-2016 | 1861 | 1861 | 1861 |
| S4-CNfix | Global | 1861-2016 | Fixed – Gridded mean-time value from S0-CNdyn | CRU-NCEP 1901-2016 | 1861-2016 | 1861-2016 | / | / | / |
| S5-CNdyn | Global | 1861-2016 | Dynamic | CRU-NCEP 1901-2016 | 1861-2016 | 1861 | 1861 | 1861 | 1861 |
| S6-CNdyn | Global | 1861-2016 | Dynamic | CRU-NCEP 1901-2016 | 1861 | 1861 | 1861 | 1861 | 1861 |
| S0-CNdyn | Global | 1861-2016 | Dynamic | CRU-NCEP recycling 1901-1920 | 1861 | 1861 | 1861 | 1861 | 1861 |



**Figures**

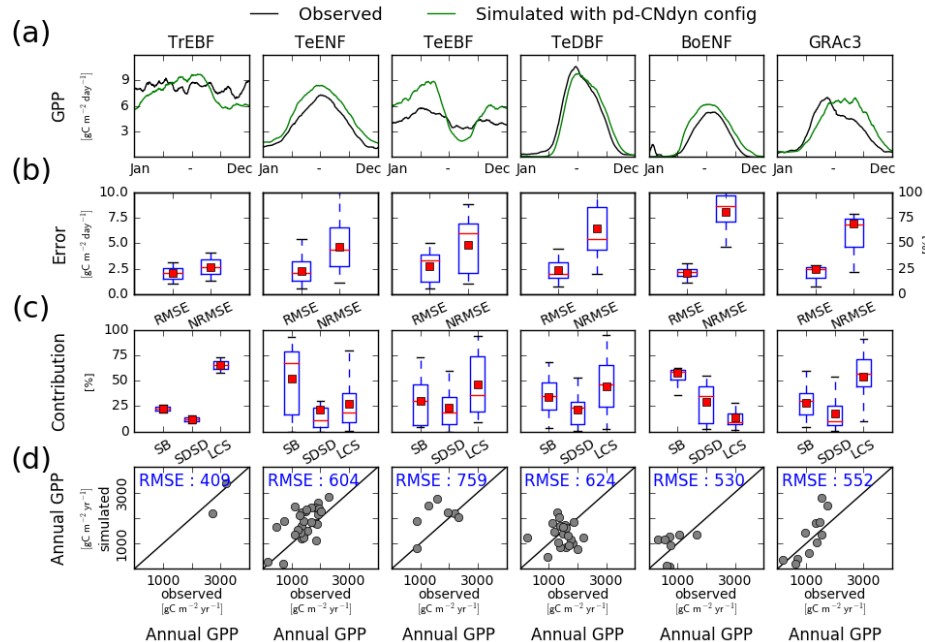

**Figure 1: Site-level evaluation of ORCHIDEE r4999 simulations against Fluxnet observations. (a) Vegetation-class mean seasonal**
5  **variations of GPP, (b) Root Mean Square Error (RMSE) and Normalized Root Mean Square Error (NRMSE) of simulated daily variations of GPP per vegetation class, (c) Attribution of the Mean Square Error (MSE) of the daily variations of GPP to model errors on mean value (SB), standard deviation (SDSD) or correlation (LCS) (Kobayashi and Salam, 2000) and (d) simulated vs. observed Annual mean GPP at site-level. On panels (b) and (c), the box extends from the lower (25 %) to upper quartile (75 %) values of the data, with a red line at the median and a red square at the arithmetic mean. The whiskers extend from the box to**
10  **show the range of the data within 1.5 × (25–75 %) data range.**



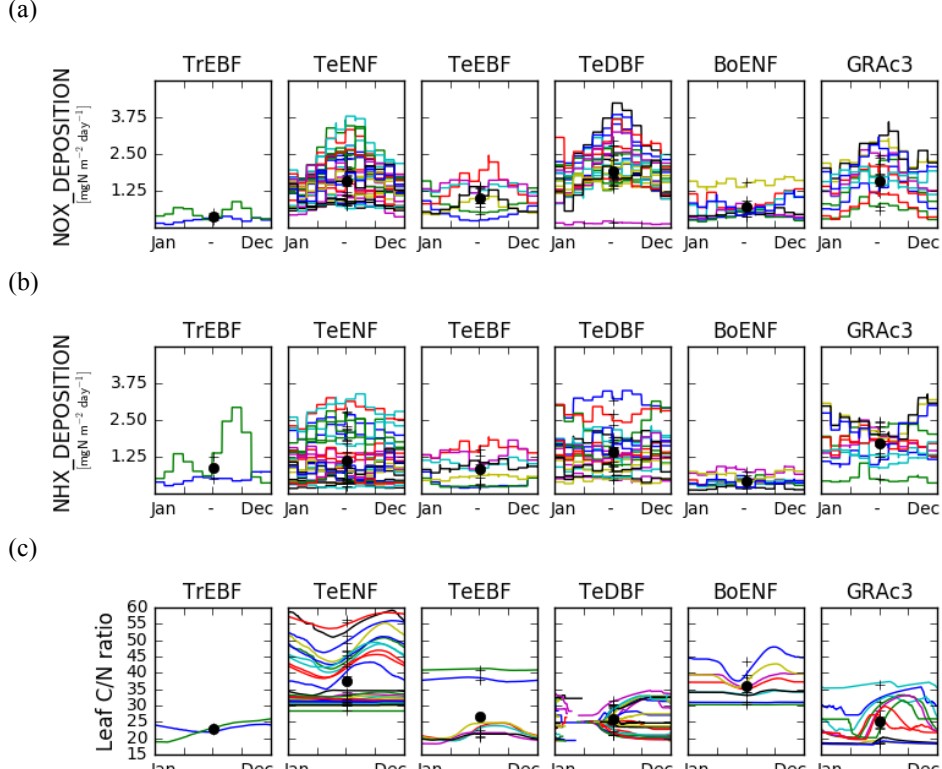

**Figure 2: Mean seasonal variations of factors driving the simulated GPP at each site per PFT class. (a) variation of the deposition of NHx compounds, (b) variation of the deposition of NOx compounds and (c) variation of the leaf C/N ratio. Each site is represented by a different colour. The black crosses denote the time-averaged value for each site and the black dots show the time-averaged and site-averaged value for each PFT. Missing data for the leaf C/N ratio of Temperate Deciduous Broadleaved Forests are due to the absence of leaves.**





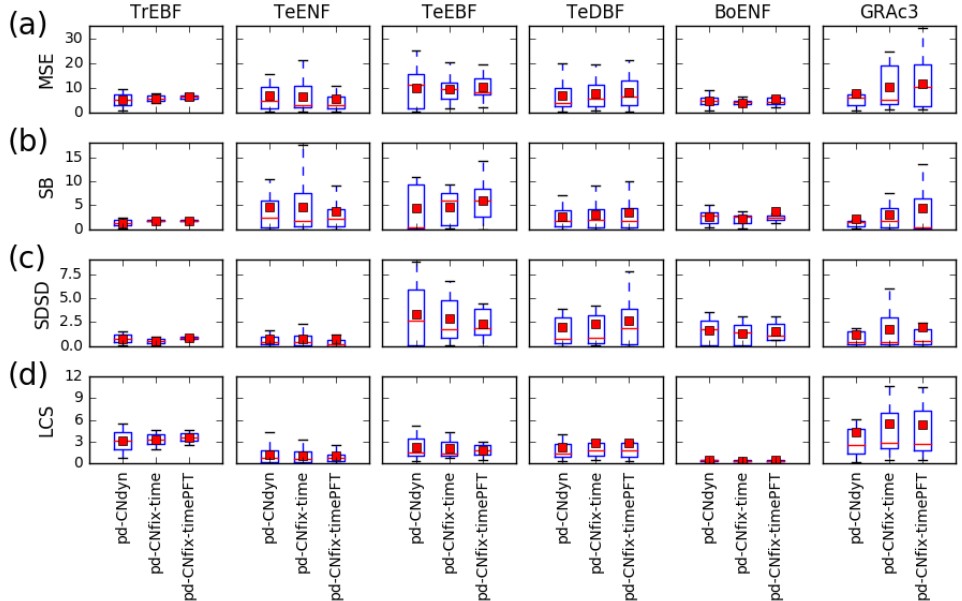

**Figure 3: Sensitivity of the model performance to the model configuration. Distribution at the plant functional type level of the Mean Square Error (MSE) of the daily GPP variations at each site and contribution to the MSE of model errors on mean value (SB), on standard deviation (SDSD) or on correlation (LCS) for the pd-CNdyn, pd-CNfix-time and pd-CNfix-timepft simulations. The box extends from the lower (25 %) to upper quartile (75 %) values of the data, with a red line at the median and a red square at the arithmetic mean. The whiskers extend from the box to show the range of the data within 1.5 × (25–75 %) data range.**




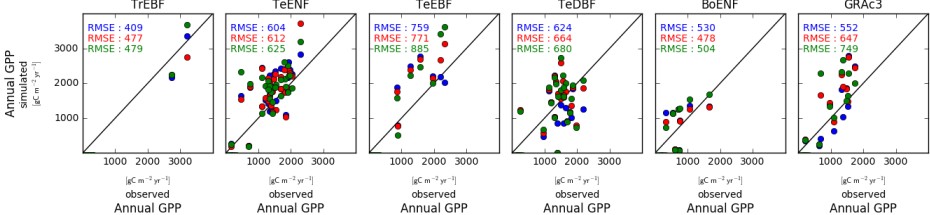

**Figure 4: Simulated vs. observed annual mean GPP at site level. Blue dots are for a model configuration in which the leaf C/N ratio varies between sites and across time (pd-CNdyn), the red dots represent a configuration in which the leaf C/N ratio varies across sites but was fixed across time (pd-CNfix-time) and the green dots represent a configuration in which the leaf C/N ratio was fixed across sites and timer (pd-CNfix-timePFT).**

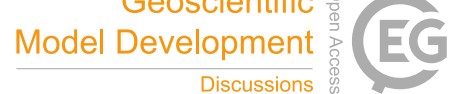



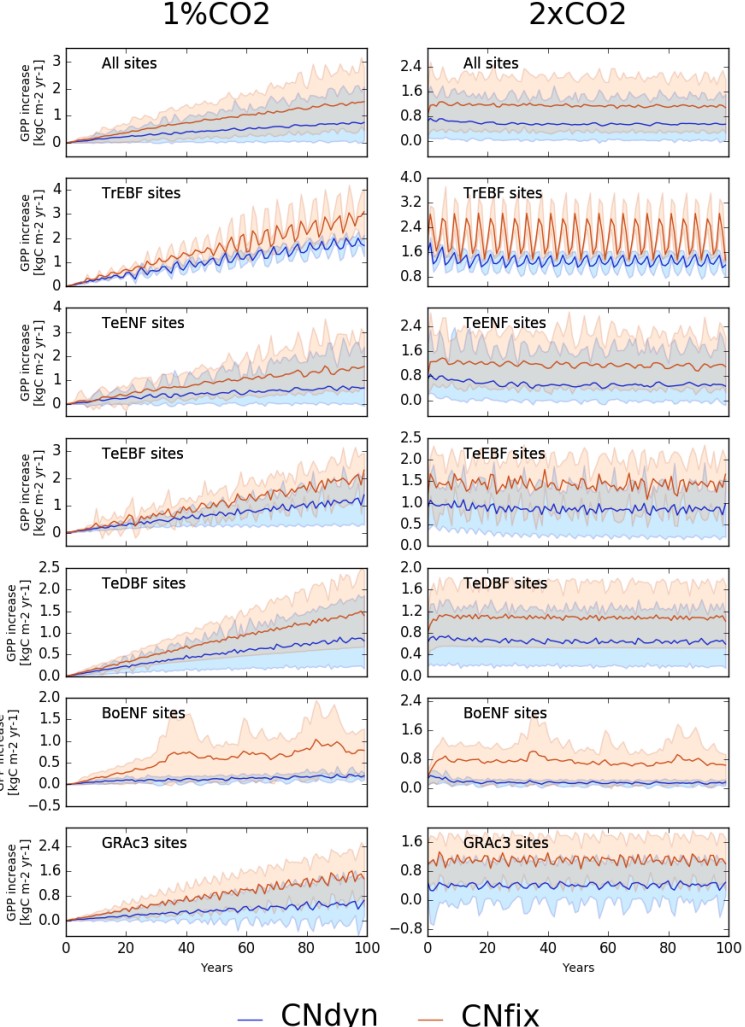

**Figure 5: Effect of changes in the atmospheric CO$_2$ concentration on GPP.** Annual mean GPP difference (kgC m$^{-2}$ yr$^{-1}$, mean, 5 and 95 percentile) between the EXP-CNdyn and pd-CNdyn simulations (in blue) and the EXP-CNfix-time and pd-CNfix-time simulations (in red) for the different PFTs, where EXP stands for 1%CO2 (left column) and 2xCO2 (right column). A relative time 5 axis was used because the simulations started the year following the last observational year in the Fluxnet database which may differ across sites.



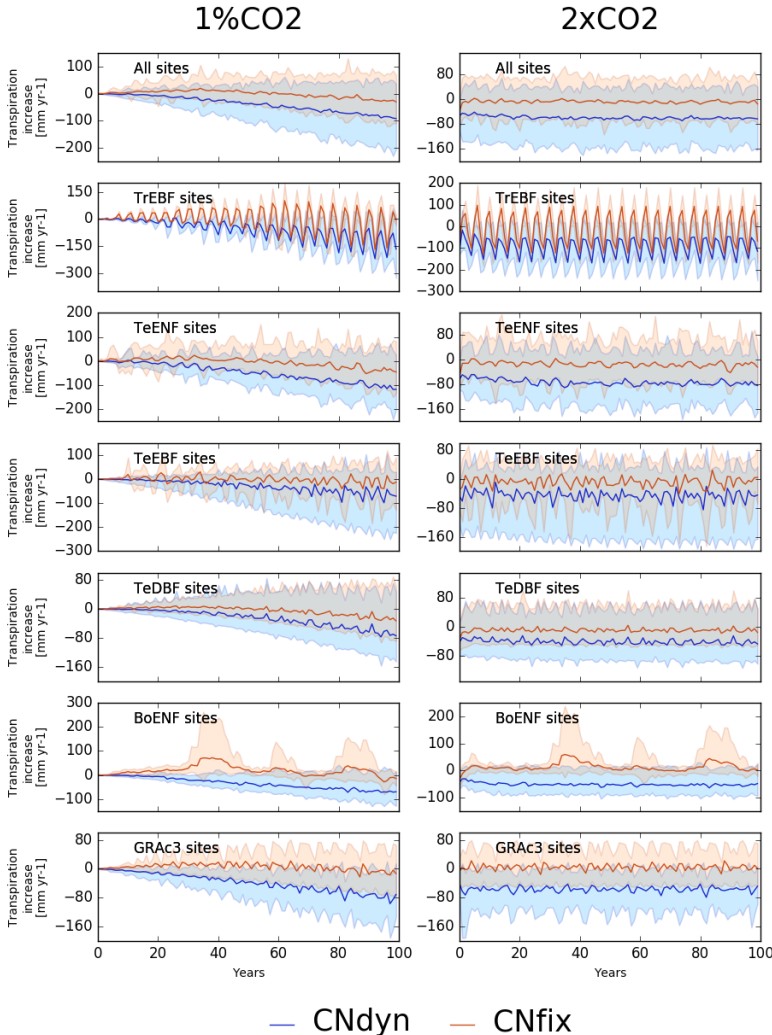

**Figure 6: Effect of changes in the atmospheric CO₂ concentration on transpiration.** Annual mean transpiration difference (kg[H2O] m⁻² yr⁻¹, mean, 5 and 95 percentile) between the EXP-CNdyn and pd-CNdyn simulations (in blue) and the EXP-CNfix-time and pd-CNfix-time simulations (in red) for the different plant functional types, where EXP stands for 1%CO2 (left column) and 2xCO2 (right column). A relative time axis was used because the simulations started the year following the last observational year in the Fluxnet database which may differ across sites.





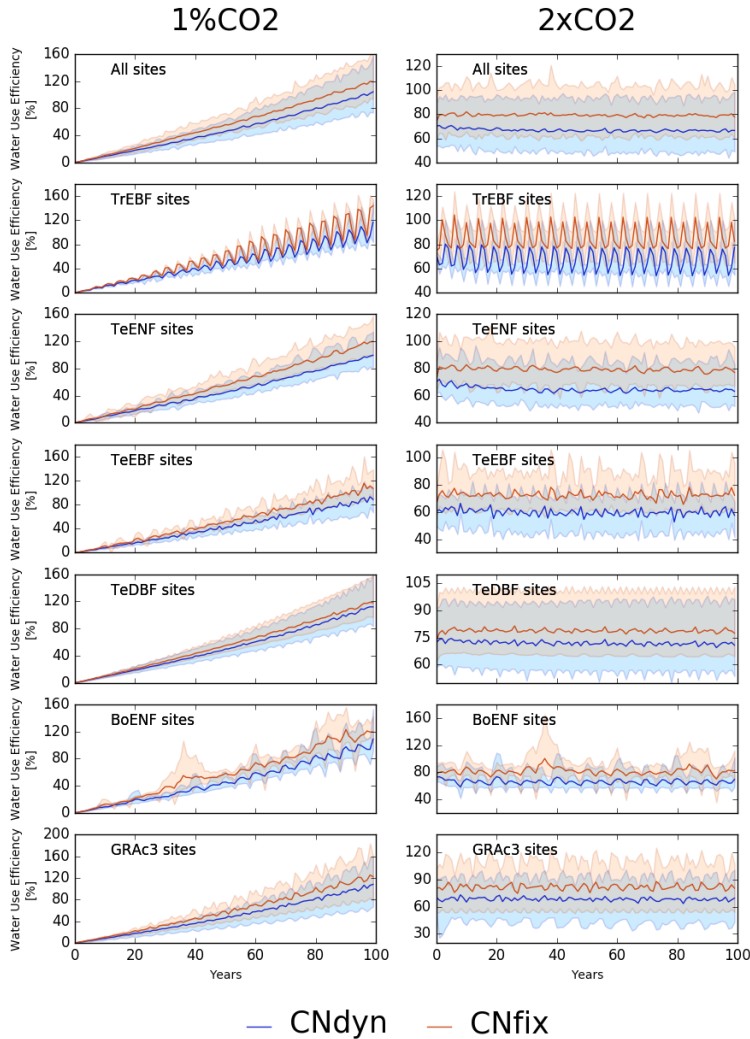

**Figure 7: Effect of changes in the atmospheric $CO_2$ concentration on water use efficiency (WUE). Annual mean WUE relative difference ([%], mean, 5 and 95 percentile) between the EXP-CNdyn and pd-CNdyn simulations (in blue) and the EXP-CNfix-time and pd-CNfix-time simulations (in red) for the different plant functional types, where EXP stands for 1%CO2 (left column) and 2xCO2 (right column). A relative time axis was used because the simulations started the year following the last observational year in the Fluxnet database which may differ across sites.**



(a)

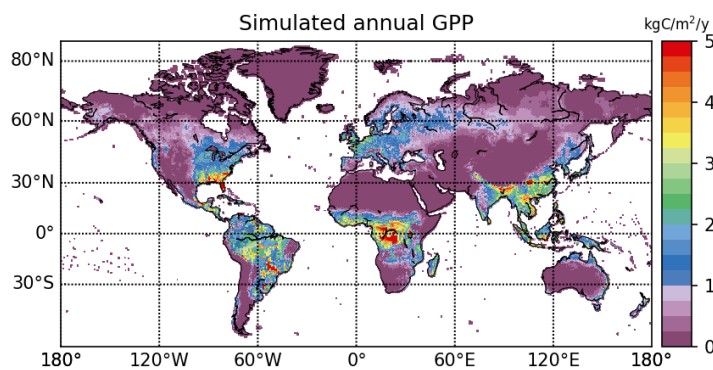

(b)

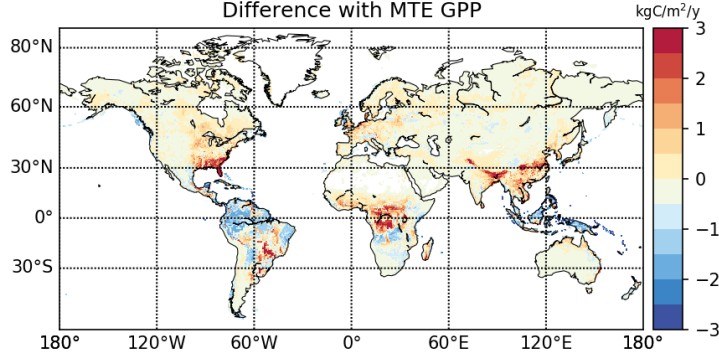

**Figure 8: Global scale evaluation of ORCHIDEE r4999 against the observation-based MTE-GPP product. (a) Global distribution of the simulated annual mean GPP (kgC m$^{-2}$ yr$^{-1}$) over 2005-2016; (b) Global distribution of the difference between the simulated annual mean GPP and the MTE-GPP product.**

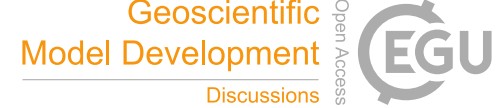


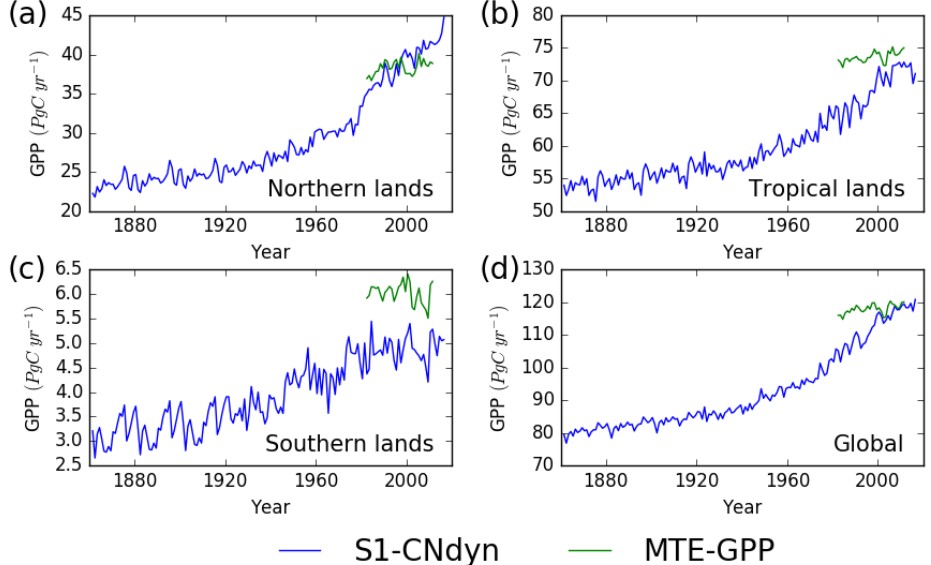

**Figure 9: Evaluation of ORCHIDEE r4999 against the observation-based MTE-GPP product for four regions. Time evolution of the annual mean GPP (PgC yr⁻¹) estimated by the ORCHIDEE model (in red) from 1860 to 2016 and by the observation-based MTE-GPP product (in green) from 1982-2008 for (a) Northern lands (>25°N), (b) Tropical lands (<25°N and >25°S), (c) Southern lands (<25°S) and (d) all lands.**



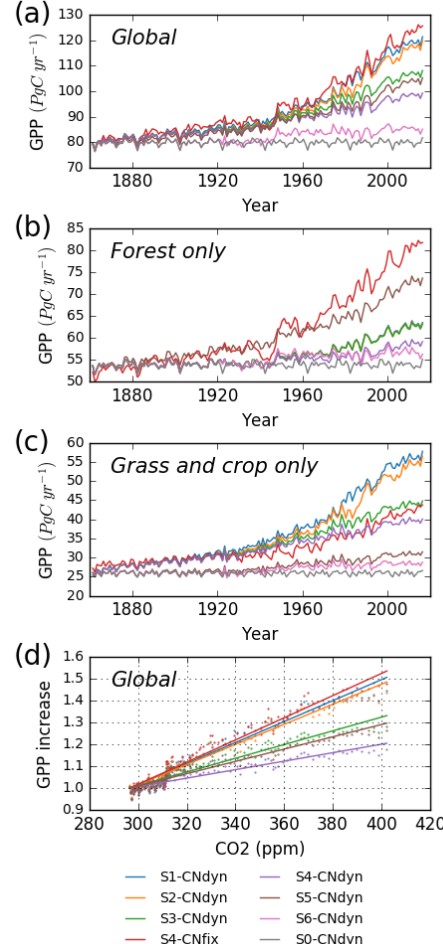

**Figure 10: Impact of key driving factors on the global annual mean GPP simulated by ORCHIDEE. (A) Simulated global annual mean GPP (PgC yr$^{-1}$) as a function of time, from 1860 to 2016; (B) Simulated annual mean GPP (PgC yr$^{-1}$) of all forest lands from**
5 **1860 to 2016; (C) Simulated annual mean GPP (PgC yr$^{-1}$) of all grasslands and croplands from 1860 to 2016; (D) Simulated global annual mean GPP (relatively to its 1900 value) as a function of the [CO$_2$] level. S0-CNdyn to S6-CNdyn are seven additive scenarios where driving factors are added one at a time, and S4-CNfix a scenario where the leaf C/N ratios of each PFT within each grid cell are fixed to their mean-time pre-industrial values (see Sect. 2.4.3 and Table 2 for details). On panel (B), blue, orange and green lines (respectively for S1-CNdy, S2-CNdyn and S3-CNdyn scenarios) overlay because synthetic fertiliser and manure**
10 **applications are not considered on forested lands and have consequently no impact on GPP.**