# Peer review of "Accounting for Carbon and Nitrogen interactions in the Global Terrestrial Ecosystem Model ORCHIDEE (trunk version, rev 4999): multi-scale evaluation of gross primary production"

_Geoscientific Model Development, 2018_

## Referee Comment (RC1) · Anonymous Referee #1 · 14 Dec 2018

Review of manuscript gmd-2018-261 "Accounting for Carbon and Nitrogen interactions in the Global Terrestrial Ecosystem Model ORCHIDEE (trunk version, rev 4999): multi-scale evaluation of gross primary production" by Nicolas Vuichard, Palmira Messina, Sebastiaan Luyssaert, Bertrand Guenet, Sönke Zaehle Josefine Ghattas, Vladislav Bastrikov, and Philippe Peylin

The manuscript describes the new version of Global Terrestrial Ecosystem Model OR-CHIDEE with Nitrogen interactions integrated into the trunk. The paper is very well written and very clearly presents the description of the nitrogen cycle and its interac-

tion with the photosynthesis and carbon/nitrogen allocation, and shows the results of the model validation and sensitivity analysis.

I think the manuscript describes significant contribution to the field of modeling of carbon and nitrogen interactions in the terrestrial biosphere, and deserves to be promptly published.

I do, however, have few minor remarks and questions, outlined below.

Page 3, line 1: typo, "pionneering" should be "pioneering"

Page 4, line 34 "Nitrogen inputs in the soil/plant system . . . (ii) biological nitrogen fixation and nitrogen fertilisation over managed grasslands and croplands. . ." I think this phrase needs some disambiguation — does it mean biological nitrogen fixation everywhere and fertilization over managed grass/crop lands, or both over managed grass/crop lands?

Page 5, lines 6-7: "Furthermore, the present study considers biological nitrogen fixation rates invariant in time and computed them as a function of evapotranspiration . . ." — if the nitrogen fixation rates are a function of evapotranspiration, and in experiments with elevated CO2 transpiration drops, does it mean that the nitrogen fixation drops as well? Based on observational evidence, is there a reason to believe that this effect is real and nitrogen fixation will drop in the elevated CO2 world? Naively, I would think that the opposite is true — more available carbon in CO2-rich world may lead to the plants being able to spent more on symbionts, increasing fixation. How does this fixation treatment effect the differences between CNfix and CNdyn experiments presented in this paper?

Page 9, lines 13-16: What is the length of the in-situ meteorological data? Is it enough to sample representative interannual variability? Under-sampling climate variability might lead to biases in the base state of the vegetation, and perhaps also to the biases in the responses to model treatment.

[Figure]

Page 16, lines 1-4, Figures 5 and 6: The enhanced interannual variability on BoENF sites in CNfix simulations (and lack of this variability in respective CNdyn) looks very interesting, especially what looks like long-term oscillations in CNfix output. What can be the cause of that, in the system with less degrees of freedom than CNdyn configuration?

Page 16, lines 26-29. What is the reason for the large GPP biases of different signs in two tropical forest regions (Africa and Amazonia)?

Page 41, line 9: typo, "S1-CNdy" should be "S1-CNdyn"

A general question: How does geographical distribution of GPP biases compare with the original ORCHIDEE model? How does it translate in the biases in other biophysical characteristics, such as biomass or LAI? I understand that the main focus of this manuscript is GPP, but I think it would be beneficial to the reader if some other results were shown too, at least from the global simulation. Unless the authors plan further publications which would address validation of the presented model version in a broader sense, of course.
* * *

---

## Referee Comment (RC2) · Anonymous Referee #2 · 26 Mar 2019

(A) General comments :

This paper describes the evaluation of a revised version of the ORCHIDEE model, incorporating representations of the carbon (C) and nitrogen (N) (and water) interactions. This paper comes 9 years after the initial publication of a first version of a C-N version of the ORCHIDEE model (Zaehle & Friend, 2010; hereafter ZF10).

As stated by the authors, this version of the ORCHIDEE model is very similar to the one already published by ZF10, with several modifications (listed from P3L30 ("Page 3

Line 30") to L4L11 and on P5L4-9). These modifications are mostly not mathematically described in the paper.

Different from ZF10 that evaluated fluxes simulated over a set of European forests, this paper provides an evaluation of the revised ORCHIDEE over GPP data acquired across the globe (using both Fluxnet data and a machine-learning product predicting GPP across the globe "MTE-GPP"). After this initial evaluation, the paper presents sensitivity analyses (SA) aiming at inferring the role of simulated C-N coupling on the centennial dynamic of simulated GPP.

When evaluating a revised version of a model, one needs two references: (1) ground-truth data and (2) a previous version of the model from which the one we are evaluating has been developed. Both are mandatory to provide a thorough evaluation of a revised version of a model, and conclude as whether or not the developments have indeed improved the model.

As regards ground-truth data:

- the model is evaluated against GPP time series. This is indeed an important flux, for which the model needs be evaluated. However, we are here dealing with a coupling of C and N cycles in the model. Evaluating the model against C flux data is clearly not enough. I know that N data are much less common than C data (e.g. Vicca et al. 2018), but the effort has already been made in earlier versions of ORCHIDEE (see ZF10 for instance). Hence I expect at least a minimal evaluation of this new version against some N data;

- the N cycle also impacts respiration. Since the Fluxnet data include both daytime and nighttime (i.e. respiration) fluxes, I see no good reason for the authors not to evaluate the model ability to simulate respiration fluxes;

- since part of the sensitivity analysis implies simulated transpiration fluxes, I also expect to see some comparison of simulated evapotranspiration against flux tower data.

[Figure]

To this respect, P3L7 is misleading stating that the paper includes a "evaluation of simulated gross carbon uptake and transpiration by plants.". I see no evaluation against transpiration data in the paper.

As regards comparison with previous versions of the model: when reading the paper, I cannot evaluate how the model modifications affected the model prediction accuracy. As said above, there are two groups of modifications listed by the authors: group 1 (p. 3-4 of the manuscript) seems to be overlooked by the authors, while group 2 (p. 5: modifications in the photosynthesis scheme and in the photosynthesis-N coupling) appear more important (i.e. the authors refer to them later in the paper). If the authors think group 2 would significantly impact the simulations, I expect to see a model comparison confronting simulations from a former (e.g. O-CN?) and the current model version. Since two main modifications are mentioned (modification of the photosynthesis scheme and modification of the photosynthesis-N coupling), I expect to see how both independently impact the model output.

Based on these two points (partial model evaluation against ground-truth data and lack of comparison with model previous versions to evaluate the impact of model modifications), I think the paper in its current version is not ready for publication.

(B) Additional comments:

P2L18, replace "is plentiful" by "is non-limiting provided adequate mineral nutrition in the future,"

P2L21, replace "will" by "would"

P3L7 "thorough"

P5L30 eq. 2: on which data were the parameters fitted ? On GPP data? These parameters are very sensitive, please be precise.

P8: How were equations 11 and 12 parameterized? Fitted on which data?

P11L15 : replace "ran" by "run" (past participle form, several occurrences throughout the text).

P11L22 "feeding back"

P14L25: "differences in the simulated mineralisation and plant Nitrogen uptake (not shown)." Is certaintly very informative (probably more that forcing Ndep time series as appears on Fig. 2), that's a pity we cannot see that.

P18L4 says that one of the modifications of the model is "the maximum Rubisco activity-limited carboxylation rate is a direct function of the leaf nitrogen content (Kattge et al., 2009)"... well that was already the case in OCN (see eq. 4 of model appendix description in ZF10).

Table 1: Where do these values come from? Parameter values are for CNleaf,min and CNleaf,max are not documented.

Fig2c,TeDBF: How does it occur that C/N either decreases or increases from June to December in TeDBF ? in NH, it should increase (leaf N decreases : N resorption while C remains about constant).

(C) References:

Vicca, S., Stocker, B. D., Reed, S., Wieder, W. R., Bahn, M., Fay, P. A., ... & Rebel, K. T. (2018). Using research networks to create the comprehensive datasets needed to assess nutrient availability as a key determinant of terrestrial carbon cycling. Environmental Research Letters, 13(12), 125006.

Zaehle, S., & Friend, A. D. (2010). Carbon and nitrogen cycle dynamics in the O‐CN land surface model: 1. Model description, site‐scale evaluation, and sensitivity to parameter estimates. Global Biogeochemical Cycles, 24(1).

---

## Author Response (AR1)

In the following, reviewers' comments are in black, whilst our responses are in red. The text added in the revised version of our manuscript is in italics.

Anonymous Referee #1

The manuscript describes the new version of Global Terrestrial Ecosystem Model OR-CHIDEE with Nitrogen interactions integrated into the trunk. The paper is very well written and very clearly presents the description of the nitrogen cycle and its interaction with the photosynthesis and carbon/nitrogen allocation, and shows the results of the model validation and sensitivity analysis. I think the manuscript describes significant contribution to the field of modeling of carbon and nitrogen interactions in the terrestrial biosphere, and deserves to be promptly published.

We thank referee #1 for their opinion on our manuscript.

I do, however, have few minor remarks and questions, outlined below.

Page 3, line 1: typo, "pionneering" should be "pioneering"
This will be corrected in the revised manuscript.

Page 4, line 34 "Nitrogen inputs in the soil/plant system...(ii) biological nitrogen fixation and nitrogen fertilisation over managed grasslands and croplands..." I think this phrase needs some disambiguation, does it mean biological nitrogen fixation everywhere and fertilization over managed grass/crop lands, or both over managed grass/crop lands?
The sentence has been rephrased as follows:
"Nitrogen inputs in the soil-plant system are related to (i) atmospheric nitrogen deposition under the form of NHx and NOy components, (ii) biological nitrogen fixation on any land category and (iii) nitrogen fertilisation over managed grasslands and croplands."

Page 5, lines 6-7: "Furthermore, the present study considers biological nitrogen fixation rates invariant in time and computed them as a function of evapotranspiration..." if the nitrogen fixation rates are a function of evapotranspiration, and in experiments with elevated $CO_2$ transpiration drops, does it mean that the nitrogen fixation drops as well? Based on observational evidence, is there a reason to believe that this effect is real and nitrogen fixation will drop in the elevated $CO_2$ world? Naively, I would think that the opposite is true : more available carbon in $CO_2$-rich world may lead to the plants being able to spent more on symbionts, increasing fixation. How does

this fixation treatment effect the differences between CNfix and CNdyn experiments presented in this paper?

Thanks for raising this issue and giving us the opportunity of clarifying what has been done. Indeed, we use a single climatology of evapotranspiration based on one reference ORCHIDEE global simulation for present-day conditions to compute the biological nitrogen fixation, for any model configuration and model simulation. Thus, biological nitrogen fixation (BNF) rates remain invariant in time but also from one model configuration to another. Because there are currently large uncertainties associated with the estimates of BNF, we think it was more suitable to assume it constant across model configurations and to analyse the modelled GPP from the different model configurations in relation with how Carbon-Nitrogen interactions are considered in each of the model configuration but not with changes on BNF. Note also that we choose to compute BNF as a function of evapotranspiration following the study of Cleveland et al. (1999) as we are still lacking for a "process-based" description of atmospheric nitrogen fixation that could be implemented in a global land surface model.

The sentence has been rephrased as follows:

*"Furthermore, in the present study, the BNF rates are computed as a function of evapotranspiration following the approach of Cleveland et al. (1999). For this purpose, a single climatology of evapotranspiration, based on a global ORCHIDEE simulation for present-day conditions, is used in all simulations performed in this study. As a consequence, the differences in modelled GPP by the different model configurations (see below) cannot be attributed to changes on BNF, an approach we consider reasonable due to the large uncertainties associated with the estimates of BNF* (Zheng et al., 2019)*."*

Page 9, lines 13-16: What is the length of the in-situ meteorological data? Is it enough to sample representative interannual variability? Under-sampling climate variability might lead to biases in the base state of the vegetation, and perhaps also to the biases in the responses to model treatment.

The length of the in-situ meteorological data varies from 1 to 16 years depending on the site. The mean length of the meteorological data is ~5 years and we agree with the referee that this is a relative short period. The relative short period for which meteorological data are available made us decide not to evaluate the model's capacity to simulate interannual variability. Hence our focus on seasonal variability, and long-term mean. Nevertheless, as raised by Referee #1, base state of the vegetation might be biased due to under-sampling of climate variability. In order to check if under-sampling may explain model bias, we plot the RMSE of the modelled daily GPP fluxes of each site against the length (in years) of measurement period (figure below). There is no clear relationship between RMSE and the length of the in-situ meteorological

data. We will address this issue in the manuscript by adding on page 9, line 16: *"The mean length of the meteorological data was 5 years and ranged between 1 to 16 years."*
And Page 14 line 7: *"RMSE was plotted against the length of the in-situ meteorological data (not shown) to check whether under-sampling of the climate variability explained part of the bias. The data did not support such a relationship (correlation -0.218, CI 95% -0.448 – 0.004) ".*

[Figure]

Figure R1-1 RMSE of the modelled daily GPP (gC m$^{-2}$ day$^{-1}$) for each site against length of the site record in years.

Page 16, lines 1-4, Figures 5 and 6: The enhanced interannual variability on BoENF sites in CNfix simulations (and lack of this variability in respective CNdyn) looks very interesting, especially what looks like long-term oscillations in CNfix output. What can be the cause of that, in the system with less degrees of freedom than CNdyn configuration?
Thanks for raising this issue that we did not investigate in the manuscript. We should first detail how the figures 5 to 7 showing annual mean difference between the EXP-CNdyn and pd-CNdyn (respectively EXP-CNfix and pd-CNfix) simulations have been built, EXP being either 1%CO2 or 2xCO2. The EXP-CNdyn (resp. EXP-CNfix) simulations start from the pd-CNdyn (resp. pd-CNfix) simulation and run for 100 years, with only the CO2 varying, the other forcing remaining constants (as in the pd-CNdyn simulation). By simplicity, we mention that the plotted differences are between EXP-CNdyn and pd-CNdyn (resp. EXP-CNfix and pd-CNfix). However, because the pd-CNdyn (resp. pd-CNfix) are not simulations at equilibrium, and in order to avoid attributing differences which are not due to the CO2 treatment, the pd-CNdyn (resp. pd-CNfix) simulations have not been simply duplicated to reach a 100-year length. Instead, the pd-CNdyn (resp. pd-CNfix) simulations have been prolonged, with all forcings fixed as in the pd-CNdyn (resp. pd-CNfix) simulations. In this way, we ensured that the differences between the EXP-CNdyn and the prolonged

pd-CNdyn (resp. EXP-CNfix and prolonged pd-CNfix) are due only to the CO2 treatment.

The long-term oscillations that the referee #1 mentions (longer than the meteorological forcing length that we recycle) are present at three sites over the eight BoENF sites; and over two sites, they are only present in the CNfix-time configuration and mainly for the control simulation (present-day CO2 concentration). Over these sites for this specific simulation (prolonged pd-CNfix-time simulations), there are periodic oscillations over periods longer than the meteorological forcing length or nearly pseudo-chaotic oscillations. Based on our model understanding, the oscillations are due to a feedback loop between GPP, water stress and leaf age (which directly impacts the maximum photosynthetic capacity of leaves and thus GPP). Sometime, there are bifurcations points where GPP starts to be lower than "expected" (based on the recycling of the meteorological forcing) due to very small water limitation. The small GPP drop leads to increase leaf age, which in turn will tend to decrease GPP and etc. This tendency may last several years (more than the meteorological forcing length) up to a point where the negative feedbacks between GPP – leaf age – water stress stops, allowing GPP to recover (See Figure R1-2). The reason why this behaviour is only exhibited in this specific simulation pd-CNfix-time appears unclear but it cannot be attributed to difference in model version or simulation setup because we use an unique model version and the same setup for all simulations (pd-CNfix-time, 2xCO2-CNfix-time, pd-CNvar-Depvar, 2xCO2-CNvar-Depvar). It is likely linked to small differences between the two configurations in the feedback between GPP, leaf age and water stress.

(a) (b)

[Figure]

Figure R1-2 (a) Annual mean GPP (gC m$^{-2}$ day$^{-1}$) for the pd-CNfix-time (red), 2xCO2-CNfix-time (black), pd-CNdyn (blue) and 2xCO2-CNdyn (green) over 100 years and (b) Leaf age (days) in the pd-CNfix-time (red), 2xCO2-CNfix-time (black), pd-CNdyn (blue) and 2xCO2-CNdyn (green) over 100 years at one Boreal Evergreen Needleleaf site

We believe that discussing this unexpected behaviour – although very interesting - goes beyond the scope of our manuscript. In order to not present differences which are not attributed to the CO2 treatment, we propose to modify the figure 5, 6 and 7, replacing only for these 3 sites, the prolonged pd-CNfix-time time series by a time series that duplicates the pd-CNfix-time time-series for 100 years. By doing so, we avoid this spurious effect that is fortunately not present in the pd-CNfix-time simulations. As an example, the modified figure 5 is here below:

[Figure]

Modified Figure 5 -

Page 16, lines 26-29. What is the reason for the large GPP biases of different signs in two tropical forest regions (Africa and Amazonia)?

The GPP biases over tropical forest regions are driven by different leaf C/N ratios across regions. The figure below represents grid-cell GPP of broadleaf

evergreen tropical forests against leaf C/N ratio and shows that GPP is highly correlated with C/N ratio. However, it is still unclear what is/are the primary drivers of the spread of the leaf C/N ratio and consequently of GPP. First analysis seems indicating that the GPP spread is partly explained by different NOx deposition rates (see figure below). Although there is certainly a combination of additional drivers, which have not been yet identified, there is no such relationship between tropical GPP and BNF, nor between GPP and NHx deposition.

(a)

[Figure]

GPP (gC m−2 day−1)

(b)

CN ratio (gC/gN)

(c)

NOx deposition (gN m−2 day−1)

Figure R1-3 Mean annual GPP (a), mean annual CN ratio (b) and mean annual NOx deposition (c) for the Tropical Broadleaf Evergreen forests in regions where annual precipitation is higher than 1300 mm per year for the period 2011-2016.

(a)                                           (b)

[Figure]

Figure R1-4 Mean annual GPP against mean annual leaf CN ratio (a) and mean annual GPP against mean annual NOx deposition (c) for the Tropical Broadleaf Evergreen forests for pixels where annual precipitation is higher than 1300 mm per year for the period 2011-2016.

We propose to add the following sentences in the discussion section to provide this information (page 19 line 5 of the initial version):
*"Nevertheless, GPP appears significantly biased – against MTE-GPP - over the tropical regions, with positive biases in Central Africa and negative ones in Amazonia. One may note that these biases are not so contrasted in the original ORCHIDEE version without the nitrogen cycle (r3977, see Figure 8c). Further analyses showed that the GPP biases over tropical forest regions are driven by different leaf C/N ratios across regions. However, it remains unclear what are the primary drivers of the spatial variation of the leaf C/N ratio and consequently of GPP. One of the drivers is likely to be NOx deposition, which is lower in Amazonia compared to Central Africa (not shown). There is no such relationship between GPP and BNF rate, nor between GPP and NHx deposition rate, in tropical regions. The drivers and/or processes that are responsible for turning large scale differences in NOx deposition into fine-scale differences in GPP are yet to be identified."*

Page 41, line 9: typo, "S1-CNdy" should be "S1-CNdyn"
This will be corrected in the revised manuscript.

A general question: How does geographical distribution of GPP biases compare with the original ORCHIDEE model? How does it translate in the biases in other biophysical characteristics, such as biomass or LAI? I understand that the main focus of this manuscript is GPP, but I think it would be beneficial to the reader if some other results were shown too, at least from the global simulation. Unless the authors plan further publications which would address validation of the presented model version in a broader sense, of course.

Thank you for the suggestion. Although our initial intention was not to focus on comparing model versions - as the models are too different to attribute different model behaviour to newly added processes. For example, the carbon allocation scheme of the ORCHIDEE model presented in this study is very different from the allocation scheme of the ORCHIDEE version without the N-cycle. Even if the current and previous versions of the model are run without accounting for the nitrogen cycle, results may differ due to differences in the carbon allocation scheme. Hence, we thought it was more interesting to specifically investigate the role of leaf C/N dynamics within the current version with the Nitrogen cycle and the C/N interactions. We realize this focus may have been a bit to narrow and, therefore, propose to update figures 8 and 9 showing the difference between modelled and observed GPP, by adding information for the original ORCHIDEE model from which we started the development of the nitrogen cycle (revision 3977 without N cycle). This will provide a comparison of how model biases have evolved owing to all developments that were necessary to implement the N-cycle, for example, a new allocation scheme.

In the revised manuscript, we also propose to show in a figure in Appendix (Figure B1) - similar to Figure 1 - the GPP model/data comparison at the site level for the former trunk version (r3977).

Although we will use this model version in future studies, we also propose to add two figures similar to figures 8 and 9 but for LAI (Figures D1 and E1). They will compare modelled LAI for S1-CNdyn simulation to the LAI provided by the GIMMS data, in terms of spatial distribution (Fig. D1) and of time evolution of the annual mean LAI for the globe and three latitudinal bands (Fig. E1). In these 2 figures, we will also add the LAI for the initial ORCHIDEE version (rev 3977).

(a)

[Figure]

(b)

[Figure]

(c)

[Figure]

Figure D1: Global scale evaluation of ORCHIDEE against the observation-based GIMMS product. (a) Global distribution of the simulated annual mean LAI by ORCHIDEE r4999 ($m^2$ $m^{-2}$) over 2001-2010; (b) Global distribution of the difference between the simulated annual mean LAI by ORCHIDEE r4999

and the GIMMS product; (c) Global distribution of the difference between the simulated annual mean LAI by ORCHIDEE r3977 and the GIMMS product

[Figure]

Figure E1: Evaluation of LAI from ORCHIDEE against the observation-based GIMMS product for four regions. Time evolution of the annual mean LAI ($m^2$ m$^{-2}$) estimated by ORCHIDEE r4999 (in blue) and ORCHIDEE r3977 (in grey) and by the observation-based GIMMS product (in green) for (a) Northern lands (>25°N), (b) Tropical lands (<25°N and >25°S), (c) Southern lands (<25°S) and (d) all lands.

The following sentences will be added on page 17 line 16 of the initial version to present the results about the LAI global distribution and mean annual values averaged per latitudinal regions:

*"Similarities between the simulated global distributions and biases in GPP and LAI (compare Fig. D1a to Fig. 8a, and Fig. D1b to Fig. 8b) suggest that the bias in GPP originates from the bias in LAI rather than from more fundamental issues with the calculation of GPP. The model/data agreement for LAI when averaged per latitudinal band is comparable to the one for GPP, with a good agreement for the Northern and Tropical lands and model underestimation in the Southern lands."*

The following sentences will be added on page 17 line 16 of the initial version to compare performances of the rev3977 and rev4999 at simulating LAI and GPP at global scale.

*"The agreement between the modelled and observed annual mean LAI and GPP summed over three latitudinal bands as well as at the global scale was*

*higher for r4999 (ie S1-CNdyn simulation) compared to r3977 without the nitrogen cycle (see Fig. 9 and E1). r3977 systematically overestimated LAI and GPP for any region, except for the Southern lands where r3977 provided similar values than the GIMMS and MTE-GPP products, respectively. Compared to GIMMS and MTE-GPP products, gridded annual mean LAI and GPP values simulated by r3977 were overestimated in the Northern lands with biases exceeding those found in r4999. On the opposite, biases of the r4999 were higher than those of r3977 in the tropical regions, in particular in Central Africa (see Fig. 8 and D1)."*

References:

Zheng, M., Zhou, Z., Luo, Y., Zhao, P. and Mo, J.: Global pattern and controls of biological nitrogen fixation under nutrient enrichment: A meta-analysis, Glob. Chang. Biol., gcb.14705, doi:10.1111/gcb.14705, 2019.

In the following, reviewers' comments are in black, whilst our responses are in red. The text added in the revised version of our manuscript is in italics.

Anonymous Referee #2

(A) General comments :This paper describes the evaluation of a revised version of the ORCHIDEE model, incorporating representations of the carbon (C) and nitrogen (N) (and water) interactions. This paper comes 9 years after the initial publication of a first version of a C-N version of the ORCHIDEE model (Zaehle & Friend, 2010; hereafter ZF10). As stated by the authors, this version of the ORCHIDEE model is very similar to the one already published by ZF10, with several modifications (listed from P3L30 ("Page 3 Line 30") to L4L11 and on P5L4-9). These modifications are mostly not mathematically described in the paper. Different from ZF10 that evaluated fluxes simulated over a set of European forests, this paper provides an evaluation of the revised ORCHIDEE over GPP data acquired across the globe (using both Fluxnet data and a machine-learning product predicting GPP across the globe "MTE-GPP"). After this initial evaluation, the paper presents sensitivity analyses (SA) aiming at inferring the role of simulated C-N coupling on the centennial dynamic of simulated GPP.

When evaluating a revised version of a model, one needs two references: (1) ground-truth data and (2) a previous version of the model from which the one we are evaluating has been developed. Both are mandatory to provide a thorough evaluation of a revised version of a model, and conclude as whether or not the developments have indeed improved the model.

We thank the reviewer for their view on model evaluation but the listed prerequisites narrow the definition of model evaluation to what is generally considered a benchmark. Contrary to the reviewer's view, the literature shows a much richer practice which reflects differences in objectives across models and model developments. For instance, looking at the temporal or spatial dynamic of a model or performing parameter sensitivity analysis are also valid ways of evaluating model behaviour and could even be more insightful than a comparison against ground truth data.

In the study under review, the objective was not to compare the former trunk version (r3977, without a nitrogen cycle) against the version presented in the manuscript and including a nitrogen cycle and the C/N interactions. The nitrogen cycle is a new functionality (compared to the former trunk version) and given its link to ecological theory we certainly want to keep it even if this implies a loss of model skill for few pools or fluxes. That's the reason why we did not focus on model evaluation against a former model version but look more precisely at the model response to the coupling/decoupling of the C and

N cycles. However, we understand that knowing how does the current model version compare to the former trunk version is an outcome of the model evaluation that could be expected by some readers. In the revised manuscript, we propose to show in an appendix (Figure B1, see below) - similar to Figure 1 - the GPP model/data comparison at the site level for the former trunk version (r3977).

[Figure]

Figure B1 : Site-level evaluation of ORCHIDEE r3977 (ie without N cycle) simulations against Fluxnet observations. (a) Vegetation-class mean seasonal variations of GPP, (b) Root Mean Square Error (RMSE) and Normalized Root Mean Square Error (NRMSE) of simulated daily variations of GPP per vegetation class, (c) Attribution of the Mean Square Error (MSE) of the daily variations of GPP to model errors on mean value (SB), standard deviation (SDSD) or correlation (LCS) (Kobayashi and Salam, 2000) and (d) simulated vs. observed Annual mean GPP at site-level. On panels (b) and (c), the box extends from the lower (25 %) to upper quartile (75 %) values of the data, with a red line at the median and a red square at the arithmetic mean. The whiskers extend from the box to show the range of the data within 1.5 × (25–75 %) data range.

Page 14 line 22 of the initial manuscript, we add the following paragraph for describing how revision 3977 (without the N cycle) compare to rev 4999 (with the N cycle) in terms of model/data agreement at Fluxnet sites:

*"In order to analyse how ORCHIDEE r4999 performs compared to r3977 (original version without the nitrogen cycle), we evaluated the GPP simulated by ORCHIDEE r3977 against Fluxnet observations (Fig. B1). The model/data agreement for r3977 was comparable to the one for r4999 but slightly better. In particular, NRMSE of the simulated daily GPP flux (Fig. B1b) and the RMSE of*

*the simulated annual mean GPP (fig. B1d) were lower in r3977 compared to r4999, for Temperate Evergreen Needleleaved and Broadleaved Forests and Temperate Deciduous Broadleaved Forest sites. Especially for Temperate Evergreen Needleleaved and Broadleaved Forests sites, the lower mean NRMSE of the simulated daily GPP at the PFT level for r3977 was due to a narrower range of NRMSE values at site level (whisker boxes are narrower), indicating that the NRMSE was not systematically lower at all sites but only at some specific ones."*

We also propose to extent figure 8 with a third panel where we map the difference between the annual mean GPP simulated by the former trunk version of ORCHIDEE (ref3977) and the annual mean GPP computed by the MTE-GPP product over 2000-2010;

Last regarding the GPP evaluation, we propose to add on Figure 9 (see below), the evolution of the global mean and latitudinal band mean GPP simulated by the former trunk version in addition to the GPP simulated by the revision 4999 (S1-CNdyn configuration) and the GPP estimated by the MTE-GPP product.

[Figure]

Figure 9: Evaluation of GPP from ORCHIDEE against the observation-based MTE-GPP product for four regions. Time evolution of the annual mean GPP (PgC yr$^{-1}$) estimated by ORCHIDEE r4999 (in blue) and ORCHIDEE r3977 (in grey) and by the observation-based MTE-GPP product (in green) for (a) Northern lands (>25°N), (b) Tropical lands (<25°N and >25°S), (c) Southern lands (<25°S) and (d) all lands.

In addition to the GPP evaluation, we now add an evaluation of the LAI at global scale (see below). Consequently, new figures have been created similar to Figures 8 and 9 but for LAI (respectively Figure D1 and E1).

(a)

[Figure]

(b)

[Figure]

(c)

[Figure]

Figure D1: Global scale evaluation of ORCHIDEE against the observation-based GIMMS product. (a) Global distribution of the simulated annual mean LAI by ORCHIDEE r4999 (m² m⁻²) over 2001-2010; (b) Global distribution of the difference between the simulated annual mean LAI by ORCHIDEE r4999

and the GIMMS product; (c) Global distribution of the difference between the simulated annual mean LAI by ORCHIDEE r3977 and the GIMMS product

[Figure]

Figure E1: Evaluation of LAI from ORCHIDEE against the observation-based GIMMS product for four regions. Time evolution of the annual mean LAI ($m^2$ $m^{-2}$) estimated by ORCHIDEE r4999 (in blue) and ORCHIDEE r3977 (in grey) and by the observation-based GIMMS product (in green) for (a) Northern lands (>25°N), (b) Tropical lands (<25°N and >25°S), (c) Southern lands (<25°S) and (d) all lands.

The following sentences have been added page 17 line 16 of the initial version to present the results about the LAI global distribution and mean annual values averaged per latitudinal regions:
*"Similarities between the simulated global distributions and biases in GPP and LAI (compare Fig. D1a to Fig. 8a, and Fig. D1b to Fig. 8b) suggest that the bias in GPP originates from the bias in LAI rather than from more fundamental issues with the calculation of GPP. The model/data agreement for LAI when averaged per latitudinal band is comparable to the one for GPP, with a good agreement for the Northern and Tropical lands and model underestimation in the Southern lands."*

The following sentences have been added page 17 line 16 of the initial version to compare performances of the rev3977 and rev4999 at simulating LAI and GPP at global scale.
*"The agreement between the modelled and observed annual mean LAI and GPP summed over three latitudinal bands as well as at the global scale was higher for r4999 (ie S1-CNdyn simulation) compared to r3977 without the*

*nitrogen cycle (see Fig. 9 and E1). r3977 systematically overestimated LAI and GPP for any region, except for the Southern lands where r3977 provided similar values than the GIMMS and MTE-GPP products, respectively. Compared to GIMMS and MTE-GPP products, gridded annual mean LAI and GPP values simulated by r3977 were overestimated in the Northern lands with biases exceeding those found in r4999. On the opposite, biases of the r4999 were higher than those of r3977 in the tropical regions, in particular in Central Africa (see Fig. 8 and D1)."*

As regards ground-truth data:
- the model is evaluated against GPP time series. This is indeed an important flux, for which the model needs be evaluated. However, we are here dealing with a coupling of C and N cycles in the model. Evaluating the model against C flux data is clearly not enough. I know that N data are much less common than C data (e.g. Vicca etal. 2018), but the effort has already been made in earlier versions of ORCHIDEE (seeZF10 for instance). Hence I expect at least a minimal evaluation of this new version against some N data;

Focusing on GPP flux was a deliberate choice, which we thought was motivated by exactly the same arguments as the reviewer:
- N data are much less common than C data. Model evaluation against such data would be mostly anecdotal (or at least very partial) which goes against the objective of a global scale model such as ORCHIDEE.
- The effort of looking at N data has already been made in an earlier version of ORCHIDEE. In this respect, we think that the model version we present here has not changed sufficient with regard to the way the N dynamics are modelled compared to ZF10 (in which the requested evaluation has been presented).

Additionally, the extensive dataset of carbon fluxes from the Fluxnet network has so far, however, not been used for evaluating any of the ORCHIDEE versions with a nitrogen cycle. Rather than reproducing ZF10 we engaged into a relatively original study of which the main findings are presented in the manuscript under review. Thus, the manuscript combines an extensive evaluation of the model GPP at FluxNet sites with an evaluation of the impact of nitrogen limitation on GPP under atmospheric CO2 increase. Such combination also directly contributes to the novelty of the manuscript.

We agree that this choice was not sufficiently motivated in the initial manuscript. Consequently, we propose to add the following sentence in the revised manuscript, Page 3 line 16 of the initial manuscript:
*"While the OCN model (Zaehle and Friend, 2010), the predecessor of ORCHIDEE r4999, has already been evaluated over a restricted set of sites for which C and N data are available, the extended C flux dataset from the*

*Fluxnet network has so far not been used for an in-depth evaluation of an ORCHIDEE version that includes the N cycle and the C/N interactions."*

- the N cycle also impacts respiration. Since the Fluxnet data include both daytime and nighttime (i.e. respiration) fluxes, I see no good reason for the authors not to evaluate the model ability to simulate respiration fluxes;

In addition to the estimate of GPP flux, the partitioning of the NEE flux measured at site provides an estimate of the Total ecosystem respiration, which includes the autotrophic respiration and the heterotrophic respiration by soil microorganisms. Because the heterotrophic respiration is highly dependent of the long-term site history in terms of land use - which we can not account for in our modelling set-up at the site level -, direct comparison of the modelled total ecosystem respiration with the one based on site measurements will include a systematic bias. For this reason, we did not evaluate the modelled ecosystem respiration against site data as neither a good nor a poor match between the data and simulation could lead to a robust conclusion concerning model performance.

- since part of the sensitivity analysis implies simulated transpiration fluxes, I also expect to see some comparison of simulated evapotranspiration against flux tower data.

In the revised manuscript, we propose to add a figure in Appendix (Figure C1) where we summarise the model/data agreement for the latent heat flux.

[Figure]

Figure C1 : Site-level evaluation of ORCHIDEE r4999 simulations against Fluxnet observations. (a) Vegetation-class mean seasonal variations of Latent

Heat flux (LE; W m$^{-2}$), (b) Root Mean Square Error (RMSE; W m$^{-2}$) and Normalized Root Mean Square Error (NRMSE; %) of simulated daily variations of LE per vegetation class, (c) Attribution of the Mean Square Error (MSE) of the daily variations of LE to model errors on mean value (SB; %), standard deviation (SDSD; %) or correlation (LCS; %) (Kobayashi and Salam, 2000) and (d) simulated vs. observed Annual mean LE at site-level (Wm$^{-2}$). On panels (b) and (c), the box extends from the lower (25 %) to upper quartile (75 %) values of the data, with a red line at the median and a red square at the arithmetic mean. The whiskers extend from the box to show the range of the data within 1.5 × (25–75 %) data range.

Page 14 line 22 of the initial manuscript, we add the following paragraph for describing the model/data agreement at fluxnet sites for the latent heat flux:
*"Because the GPP flux is intimately linked to the transpiration flux through stomatal control, a site-level evaluation of the pd-CNdyn simulation has been performed against site-level observations of the latent heat (LE) flux (Fig. C1), an energy flux to which transpiration contributes to, as does the soil evaporation. Overall, the model performed better at simulating LE variations than variations in GPP. This was particularly true when looking at the NRMSE of the simulated daily flux, which never exceeded 50% as a mean average score at the PFT level for LE, while it went to values up to 75% for GPP for some PFTs (BoENF and GRAc3)."*

To this respect, P3L7 is misleading stating that the paper includes a "evaluation of simulated gross carbon uptake and transpiration by plants.". I see no evaluation against transpiration data in the paper.
Although we now add a model evaluation for the the latent heat flux in the manuscript, we prefer removing "transpiration" and keeping only "gross carbon uptake", as it is the key variable we are focus on, in the manuscript. Thus, we rephrased as followed:
*"evaluation of simulated gross carbon uptake by plants."*

As regards comparison with previous versions of the model: when reading the paper, I cannot evaluate how the model modifications affected the model prediction accuracy. As said above, there are two groups of modifications listed by the authors: group 1(p. 3-4 of the manuscript) seems to be overlooked by the authors, while group 2 (p.5: modifications in the photosynthesis scheme and in the photosynthesis-N coupling) appear more important (i.e. the authors refer to them later in the paper). If the authors think group 2 would significantly impact the simulations, I expect to see a model comparison confronting simulations from a former (e.g. O-CN?) and the current model version. Since two main modifications are mentioned (modification of the photosynthesis scheme and modification of the

photosynthesis-N coupling), I expect to see how both independently impact the model output.

Based on these two points (partial model evaluation against ground-truth data and lack of comparison with model previous versions to evaluate the impact of model modifications), I think the paper in its current version is not ready for publication.

We understood this point that is the summary of the concerns detailed by the reviewer. We hope that we were able to discuss the referee's concerns point-by-point and list the main proposed changes:
- Including model vs data evaluation for GPP, LAI and LE through in-text changes and additional figures in Appendix
- Including model vs model evaluation for GPP, and LAI through in-text changes and additional figures in Appendix

(B) Additional comments:

P2L18, replace "is plentiful" by "is non-limiting provided adequate mineral nutrition in the future,"
We thank the reviewer for this suggestion but modifying the sentence as proposed will be in conflict with the rest of sentence where we state that "it remains questionable whether sufficient nutrients, in particular nitrogen, will be available". We propose the following change:
"Even if atmospheric [$CO_2$] will be plentiful in the future, it remains questionable whether sufficient nutrients, in particular nitrogen, will be available to fully sustain the increase of primary production associated solely to the rise of [$CO_2$]."

P2L21, replace "will" by "would"
This will be corrected in the revised manuscript.

P3L7 "thorough"
This will be corrected in the revised manuscript.

P5L30 eq. 2: on which data were the parameters fitted ? On GPP data? These parameters are very sensitive, please be precise.
We will give more details in the revised manuscript. We propose to modify the sentence as follows:
"where $a_{rJ,V}$ and $b_{rJ,V}$ are fitted parameters of the relationship between observation-based values of $J_{max,ref}/Vc_{max,ref}$ and $t_{growth}$, and equal to 2.59 [-] and -0.035 [$°C^{-1}$], respectively"

P8: How were equations 11 and 12 parameterized? Fitted on which data?

Equations 11 and 12 are empirical functions adapted from Zaehle and Friend (2010), whom parameters have been adjusted to match the shape of the equation 21 
[revised manuscript text omitted]

---

## Author Response (AR2)

In the following, reviewers' comments are in black, whilst our responses are in red. The text added in the revised version of our manuscript is in italics. Page and line numbers refer to the former version of the manuscript.

5   Anonymous Referee #3

This is a well-written paper which documents important developments to the ORCHIDEE model. I have a few minor comments for clarification.

10   Transpiration: The N limitation reduces GPP and reduces water loss through transpiration. This is an important impact of the developments in ORCHIDEE, and it is mentioned throughout the manuscript. How do these results fit within current understanding of the linked hydrologic cycle responses to a coupled CN cycle?
At the moment this is not mentioned in the Introduction but I think it should be.
15   In order to mention current knowledge on the link between hydrologic cycle, carbon and nitrogen cycle, we added in the revised manuscript the following sentence in the Introduction section (page 2 line 20):
*"Several observation-based and modelling studies highlighted the tight interactions between [$CO_2$] level, nitrogen and water resources (Felzer et al., 2009, 2011; McCarthy et al., 2010; Reich et al.,*
20   *2014) and how water and nitrogen availability may limit the [$CO_2$] fertilization effect (McCarthy et al., 2010; Reich et al., 2014), although elevated [$CO_2$] has also the capacity of improving plant water-use efficiency (Conley et al., 2001; Drake et al., 1997)."*

Also, what equation is used to relate stomatal conductance to CO2? This is relevant since it will
25   affect the strength of this transpiration/gs effect.
Method section from page 5 line 21 to page 5 line 29 has been rephrased and extended in order to include information on stomatal conductance modelling:
*"Yin and Struik (Yin and Struik, 2009) propose a C4-equivalent version of the FvCB model and analytical solutions to the set of equations which link the net assimilation rate (A, $\mu mol\ CO_2\ m^{-2}_{[leaf]}\ s^{-1}$*
30   *$^{1}$), the stomatal conductance to $CO_2$ ($g_{s[CO2]}$, $mol\ CO_2\ m^{-2}\ s^{-1}$), and the intercellular $CO_2$ partial pressure ($C_i$, $\mu mol\ mol^{-1}$) . ORCHIDEE r3977 and r4999 retained most of the formulations and parameterizations of the FvCB model as proposed by Yin and Struik (2009) except for the maximum rate of Rubisco activity-limited carboxylation ($Vc_{max}$, $\mu mol\ CO_2\ m^{-2}_{[leaf]}\ s^{-1}$) and the maximum rate of electron transport ($e^-$) transport under saturated light ($J_{max}$, $\mu mol\ e$- $m^{-2}_{[leaf]}\ s^{-1}$) for C3 plants (see*
35   *below).*
*Stomatal conductance is coupled to leaf photosynthesis and is defined as:*

$$g_{s[CO2]} = g_0 + \frac{A+R_d}{C_i - C_{i*}} f_{VPD},$$   *(1)*

*where g0 is the residual stomatal conductance if the irradiance approaches zero, $C_{i*}$ is the $C_i$-based $CO_2$ compensation point in the absence of $R_d$. $f_{VPD}$ is a coupling factor between A and $g_s$ function of*
40   *leaf-to-air vapour pressure difference that we approximate by the air vapor pressure deficit (VPD, kPa):*

$$f_{VPD} = \frac{1}{1/(a_1 - b_1 VPD) - 1},$$ (2)

*where $a_1$ and $b_1$ are empirical constants equal to 0.85 (-) and 0.14 kPa$^{-1}$, respectively.*
*In ORCHIDEE r4999, $Vc_{max}$ and $J_{max}$ are based on the formulations proposed by Kattge and Knorr (2007). $Vc_{max}$ and $J_{max}$ at temperature T, are defined as:"*

Model description:
In line with some of the previous reviewers' comments, I still found it a bit confusing to decipher which versions of the model and which developments are relevant in this study. Although it is the basis for r4999, this manuscript does not assess OCN (which is fine), but which developments mentioned in the Methods are in both versions of the model evaluated here? I am thinking in particular of the 4 changes listed at the top of page 4, the developments in 2.1.1 and 2.1.2. Perhaps a brief summary at the end of the model description section would help (and would also give a chance to introduce r3977 before the results section).

Thanks for the suggestion. We revised the Methods section in order to clarify the changes that have been performed in the two versions used in our study. These revisions consist essentially in naming ORCHIDEE r4999 and ORCHIDEE r3977 where needed.
Page 4 line 2: "The main changes *which are not directly related to the computation of the GPP* consist of"
Page 4 line 16: "*The changes cited above are in ORCHIDEE r4999 and ORCHIDEE r3977, a version comparable to r4999 but without the nitrogen cycle and the carbon-nitrogen interactions that is used in this study as a benchmark reference."*
Page 4 line 17: The developments *performed specifically* in ORCHIDEE r4999
Page 5 line 8: "*Two main modifications compared to the OCN model (Zaehle and Friend, 2010) related to the carbon assimilation or photosynthesis scheme (Yin and Struik, 2009) and refinements of the N-dependency of the photosynthetic activity (Kattge et al., 2009) have also been performed."*
instead of "The main modifications compared to the work of Zaehle and Friend (2010) are related to the carbon assimilation or photosynthesis scheme (Yin and Struik, 2009) and refinements of the N-dependency of the photosynthetic activity (Kattge et al., 2009)."
Page 5 line 18: "The updated carbon assimilation scheme used in ORCHIDEE *r3977 and* r4999 has been proposed by Yin and Struik (2009)."
Page 5 line 25: "*ORCHIDEE r3977 and r4999 retained most of the parameter values of the FvCB model as proposed by Yin and Struik (2009), except the parameters which define the maximum rate of Rubisco activity-limited carboxylation ($Vc_{max}$, µmol $CO_2$ m$^{-2}$$_{[leaf]}$ s$^{-1}$) and the maximum rate of electron transport ($e^-$) transport under saturated light ($J_{max}$, µmol e- m$^{-2}$$_{[leaf]}$ s$^{-1}$) for C3 plants."* and
"*In ORCHIDEE r3977 and r4999, $Vc_{max}$ and $J_{max}$ were based on the formulations proposed by Kattge and Knorr (2007). $Vc_{max}$ and $J_{max}$ at temperature T, are defined as:"* instead of "ORCHIDEE r4999 retained most of the parameter values of the FvCB model as proposed by Yin and Struik (2009), except the parameters which define the maximum rate of Rubisco activity-limited carboxylation

($Vc_{max}$, µmol $CO_2$ $m^{-2}_{[leaf]}$ $s^{-1}$) and the maximum rate of electron transport ($e^-$) transport under saturated light ($J_{max}$, µmol $e-$ $m^{-2}_{[leaf]}$ $s^{-1}$) for C3 plants, which were replaced by the formulation and parameterization proposed by Kattge and Knorr (2007). In ORCHIDEE (r4999), $Vc_{max}$ and $J_{max}$ at temperature $T$, are defined as:"

Page 6 line 12: "*In ORCHIDEE r3977 (like in the former versions of ORCHIDEE), the photosynthetic activity is independent of the leaf nitrogen content. As a consequence, the value of $Vc_{max,ref}$ at top of canopy is a fixed parameter. From this value, $Vc_{max,ref}$ along the canopy profile decreases exponentially, to reflect known leaf nitrogen content decrease. On the opposite,* ORCHIDEE r4999"

Page 7 line 7: "Two model configurations were developed *for ORCHIDEE r4999* to allow a straightforward analysis of the effect of the nitrogen cycle on plant productivity"

Also, in order to better structure the Methods section, we propose:
- to move the first paragraph of the Methods section at the end of the Introduction section, because it presents the main objectives of the study in terms of model evaluation: *"The evaluation is focused on the added value of including the nitrogen cycle for the purpose of simulating gross carbon uptake fluxes, including also the impact on the related plant transpiration. The evaluation consists of: (1) site-level simulations in order to assess the overall performance of the ORCHIDEE model at simulating GPP flux at Fluxnet stations (annual mean value, seasonal variations, site-to-site differences); (2) sensitivity tests to quantify the contributions of accounting for seasonal and site-to-site variations of the leaf C/N ratio to the simulated seasonal variations and mean annual GPP; (3) idealized simulations to quantify the impact of N-limitation on GPP under [CO$_2$] enrichment scenarios; and (4) global simulations in order to evaluate and analyse seasonal, long-term variations and global distribution of the simulated GPP. While the OCN model (Zaehle and Friend, 2010), the predecessor of ORCHIDEE r4999, has already been evaluated over a restricted set of sites for which C and N data are available, the extended C flux dataset from the Fluxnet network has so far not been used for an in-depth evaluation of an ORCHIDEE version that includes the N cycle and the carbon-nitrogen interactions."*
- to move the paragraph detailing the computation of BNF (Page 5 line 10) where BNF is introduced (Page 5 line 5): *"In ORCHIDEE r4999, the BNF rates are computed as a function of evapotranspiration following the approach of Cleveland et al. (1999). In the present study, a single climatology of evapotranspiration, based on a global ORCHIDEE simulation for present-day conditions, is used in all simulations performed. As a consequence, the differences in modelled GPP by the different model configurations (see below) cannot be attributed to changes on BNF, an approach we consider reasonable due to the large uncertainties associated with the estimates of BNF (Zheng et al., 2019). This approach is specific to the present study and does not preclude of using an online computation of BNF based on time-varying evapotranspiration in future studies. The forcings used for the other nitrogen input fields are detailed in sections 2.3.4 and 2.3.5."*

BNF: The approach for BNF in this study (using a single climatology of ET from a control run) does seem reasonable but how will it be handled for other applications of this model (such as being run in a coupled mode) – is there a method for calculating a climatology of ET as the simulation evolves, or will it always rely on the same climatology? (Page 5, lines 10-15)

The approach of using a single climatology for all the simulations is specific to the present study. It does not preclude of using an online computation of BNF based on time-varying evapotranspiration in future studies. This has been added in the revised manuscript (page 5, line 15):

*"This approach is specific to the present study and does not preclude of using an online computation of BNF based on time-varying evapotranspiration in future studies."*

Results and Discussion:

Page 18, Paragraphs from Line 20-32: Isn't it possible that the biases in GPP cause the biases in LAI, rather than the other way around?

It is correct that the interaction between GPP and LAI is 2-ways and that we cannot ensure that the model deficiency is first a bias on LAI which translates into a bias on GPP and not the opposite. However, we think that the link from LAI to GPP is more direct and constrained as it mainly consists in an up-scaling of the computation of GPP from leaf to canopy level, while the computation of the LAI is related to the computation of GPP (the creation of assimilates) but also to the way that the assimilates are allocated into the different biomass pools. The way that allocation is modelled may be potentially deficient and improvements of the allocation parameterisation may directly translate into improvement on GPP.

We rephrase the sentence page 18 line 20, in order to be less affirmative:

*"Similarities between the simulated global distributions and biases in GPP and LAI (compare Fig. D1a to Fig. 8a, and Fig. D1b to Fig. 8b) suggest that a GPP bias reduction may potentially be obtained from LAI modelling improvements."*

It was good to see a discussion of GPP might be so much lower in tropical Amazon and Indonesia in r4999 in the Discussion. However GPP biases also appear lower in high latitudes (the overestimation is removed) – is the reason for this improvement understood? Does it have to do with plants being unable to access N in frozen soils, or is there something else going on?

Yes, GPP in high latitudes is driven by nitrogen limitation through slow SOM decomposition and temperature effect. In high latitudes, photosynthetic capacity (ie. Vcmax values) are comparable between r3977 and r4999, but r4999 may support lower LAI values compared to r3977 due to the N limitation.

The explanation of the large effect of dynamic CN for C3 grasses has not really explained why they are so different: What is meant by "extreme" conditions which lead to the depletion of the labile pool of N. Does this occur during certain climatic events such as droughts or heat waves? Can anything more specific be said about why the pool becomes depleted at these sites but not at any of the others? (Page 21, Paragraph starting from Line 20)

The adjective "extreme" was referring to the nitrogen availability and nitrogen uptake, not to the climate conditions. They are conditions in which nitrogen availability and uptake are extremely low. First analysis indicates that the reduction of the nitrogen uptake is primary due to a low fine root biomass rather than a temperature or soil moisture effect on N uptake. We propose to rephrase these sentences as follows:
"These extreme conditions *in terms of nitrogen availability* and their consequences on the biomass allocation cannot be captured with the CN-fix configuration and it appears that at many C3 grassland sites, these extreme conditions are sufficiently frequent to produce different model responses and performances between the pd-CNfix-time and pd-CNfix-timePFT simulations and the pd-CNdyn simulation. *First analysis indicates that the low nitrogen availability is due to a reduction of the nitrogen uptake which is primary explained by the low fine root biomass rather than a temperature or soil moisture effect on the N uptake.*"

Minor comments:
Page 14, Line 27-31: Should this be MSE or MSD? It is MSD in Eq. 13.
As we refer to a deviation of observed fluxes, we prefer to use the wordings Mean Squared Error and Root Mean Squared Error, rather than Mean Squared Deviation and Root Mean Squared Deviation. RMSD and MSD have been replaced by RMSE and MSE in the revised manuscript.

Table 2 is very helpful but reference it earlier, ie in Section 2.4.2
This has been done in the revised manuscript.

I think it's relevant to point out earlier in the manuscript that ORCHIDEE with these CN developments are being used in the CMIP6 experiments. This increases the importance of understanding the large-scale effects of the CN interactions.
We have added a sentence at the end of the introduction in the revised manuscript referring to the CMIP6 experiments:
*"The resulting ORCHIDEE version developed (r4999) will become soon the standard ORCHIDEE planned to be used in the phase 6 of CMIP in an alternative version of the IPSL-CM6 model."*

Also, although not necessary I wonder if the model needs a name, could this be OCN2?? Referring to ORCHIDEE r4999 seems bulky for future reference, if this will now replace OCN in future model comparison projects.

OCN is used and developed out of the group developing the ORCHIDEE model, so ORCHIDEE r4999 won't replace OCN model. The version presented in this manuscript will become the standard ORCHIDEE version, but we prefer to keep ORCHIDEE r4999 in the manuscript, as GMD requests for referring to a specific revision number.

Page 18 Lines 9-12 There is some repetition of information here.
The sentences have been rephrased as follows:

[revised manuscript text omitted]